# Accelerated On-Device Forward Neural Network Training with Module-Wise Descending Asynchronism

**Xiaohan Zhao**[1]
xiaotuzxh@gmail.com

**Hualin Zhang**[2]
zhanghualin98@gmail.com

**Zhouyuan Huo**[*]
huozhouyuan@gmail.com

**Bin Gu**[2,3†]
jsbugin@gmail.com

[1] Nanjing University of Information Science and Technology, China
[2] Mohamed bin Zayed University of Artificial Intelligence, UAE
[3] School of Artificial Intelligence, Jilin University, China

## Abstract

On-device learning faces memory constraints when optimizing or fine-tuning on edge devices with limited resources. Current techniques for training deep models on edge devices rely heavily on backpropagation. However, its high memory usage calls for a reassessment of its dominance. In this paper, we propose forward gradient descent (FGD) as a potential solution to overcome the memory capacity limitation in on-device learning. However, FGD's dependencies across layers hinder parallel computation and can lead to inefficient resource utilization. To mitigate this limitation, we propose AsyncFGD, an asynchronous framework that decouples dependencies, utilizes module-wise stale parameters, and maximizes parallel computation. We demonstrate its convergence to critical points through rigorous theoretical analysis. Empirical evaluations conducted on NVIDIA's AGX Orin, a popular embedded device, show that AsyncFGD reduces memory consumption and enhances hardware efficiency, offering a novel approach to on-device learning.

## 1 Introduction

Deep learning models have increasingly gained attraction in a multitude of applications, showcasing exceptional predictive capabilities. Nevertheless, their rapidly expanding size [19] poses a formidable challenge for resource-limited edge devices, such as mobile phones and embedded systems. These devices are pervasive in our society and continuously generate new data. To attain model customization, user privacy, and low latency, these devices necessitate on-device learning, involving training and fine-tuning models on freshly gathered data [46]. However, the restricted memory capacity of these devices emerges as a significant hindrance. For example, the Raspberry Pi Model A, introduced in 2013, only featured 256 MB of memory [17], while the more recent Raspberry Pi 400, released in 2020, modestly increased this capacity to a mere 4 GB.

Various techniques have been proposed to address this issue, encompassing quantized training, efficient transfer learning, and rematerialization. Quantized training curtails memory consumption by utilizing low-precision network representation [22, 15, 42, 43, 8]. Efficient transfer learning diminishes training costs by updating merely a subset of the model [20, 5, 2, 44]. Lastly, rematerialization

---

[*]Author Zhouyuan Huo is currently at Google. No work performed at Google.
[†]Corresponding author.

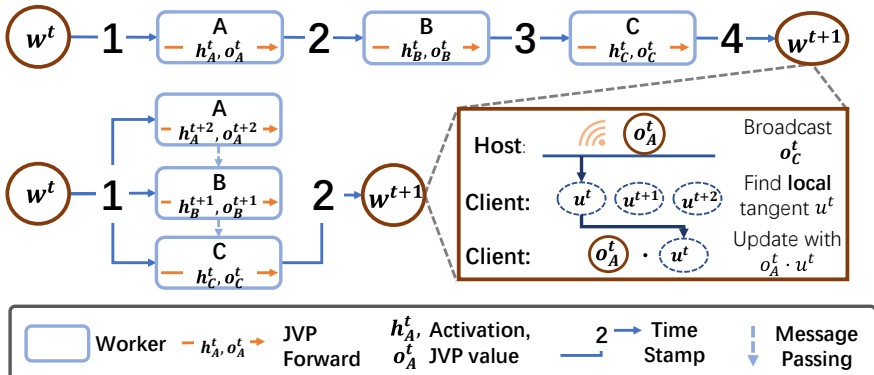

Figure 1: Comparison of Vanilla FGD and AsyncFGD, where A, B, and C signify workers in the system. Through the allocation of tasks from varying iterations, AsyncFGD breaks forward locking in FGD, thereby maximizing worker utilization.

conserves memory at the expense of time by discarding and recomputing intermediate variables [3, 7, 16, 35].

Although most of these techniques can be applied irrespective of the optimization strategy, they are frequently employed in tandem with backpropagation in deep learning. Due to the reciprocal nature of backpropagation (i.e., activations from the forward pass are preserved for the subsequent backward pass), the training of deep models utilizing backpropagation typically commands a memory footprint that is 3-4× larger than the number of parameters involved. Consequently, reexamining backpropagation within the context of surpassing the memory capacity barrier is crucial due to its elevated memory consumption.

The recent revival of interest in the forward-forward algorithm proposed by [10], along with other algorithms predicated on forward computation [39, 28, 1, 33], has prompted a reevaluation of backpropagation. The question persists: is it essential to store intermediate variables and pause to propagate gradients? An alternative, forward gradient descent (FGD), leverages the Jacobian-vector product (JVP) under automatic differentiation [37, 1, 33] to assess the effect of a stochastic perturbation in conjunction with the forward pass, yielding an unbiased approximation of the true gradient. FGD offers benefits in terms of memory consumption and biological plausibility[31], as it solely employs forward passes and substantially reduces the storage of intermediate variables [31]. Moreover, FGD can be combined with other existing techniques, such as quantized training and efficient transfer learning, potentially further diminishing memory costs.

Nonetheless, the intrinsic locking mechanism within FGD, more specifically, the layer-by-layer sequential computation during the forward pass, poses an impediment to parallel computation across disparate layers as shown in Fig. 1. In response to these challenges, this paper aims at training deep models on edge devices with memory constraints from an optimization perspective, while improving resource utilization by breaking the lock inside the forward pass with provable convergence. Thus, we propose AsyncFGD, an asynchronous version of Forward Gradient Descent with module-wise asynchronism in the forward pass to disentangle its dependencies, allowing simultaneous computation on different workers. The induced module-wise decreasing and bounded staleness in the parameters not only accelerates prediction and training but also provides theoretical guarantees in convergence. We empirically validate our method across multiple architectures and devices, including CPUs, GPUs, and embedded devices. Our results indicate that AsyncFGD reduces memory consumption and improves hardware efficiency, achieving efficient training on edge devices.

## 2 Related Works

### 2.1 Forward Gradient with Reinforcement Learning

This research builds upon the concept of forward-mode automatic differentiation (AD) initially introduced by [37]. It has since been applied to learning recurrent neural networks [39] and calculating

Hessian vector products [28]. However, exact gradient computation using forward-mode AD requires the complete and computationally expensive Jacobian matrix. Recently, Baydin et al. [1] and Silver et al. [31] proposed an innovative technique for weight updating based on directional gradients along randomly perturbed directions. These algorithms have connections to both reinforcement learning (RL) and evolution strategies (ES), since the network receives global rewards in each instance. RL and ES have been successfully utilized in specific continuous control and decision-making tasks in neural networks [38, 34, 32, 4]. Clark et al. [4] observed that global credit assignment performs well in vector neural networks with weights between vectorized neuron groups. However, in forward-model AD methods, much like in evolution strategies, the random sampling approach can lead to high variance in the estimated gradient, particularly when optimizing over a large-dimensional parameter space [31].

## 2.2 Parallel Strategies and Asynchronous Approaches

Various parallel frameworks have been developed to accelerate computations by utilizing multiple workers. These frameworks include data parallelism [21], pipeline parallelism [12, 30, 23, 14], and tensor parallelism [29, 24]. Each approach leverages different dimensions of data - batch dimension for data parallelism, layer dimension for pipeline parallelism, and feature dimension for tensor parallelism - to improve computational efficiency.

However, in edge computing environments, data is often collected and processed frame-by-frame rather than in batches. This is particularly true for scenarios requiring real-time adaptation and streaming data learning. In such contexts, edge models must quickly adapt to newly acquired data in real-time, leaving limited scope for collecting, fetching, and batching historical data. Therefore, algorithms that do not depend on batch size are preferable.The ability to align with this pipeline and train the model on the fly with incoming data is crucial. Therefore, our research specifically focuses on pipeline parallelism.

Pipeline parallelism can be categorized into synchronous and asynchronous pipelines, which divide computations into sequential stages and align well with the data stream on edge devices. However, in edge devices with limited computational power, potential idle workers due to synchronization in synchronous pipelines like [12] are not optimal.

Asynchronous parallelism, which allows non-simultaneous task execution to enhance resource utilization, is particularly relevant to our method. [23, 14] parallelize tasks by executing them from different iterations. However, the additional memory overhead of storing replicate copies to handle staleness in [23] and the use of multiple activations for backpropagation make the algorithm memory inefficient. This memory overhead is mitigated in strategies such as "rematerialization" [40, 13] and weight estimation [41]. However, these strategies were originally designed for backpropagation, while our focus is on FGD, which has distinct computational characteristics, rendering the existing "1B1F" strategy for work scheduling in [40, 13, 41, 23] unsuitable.

## 3 Preliminaries

**Gradient-based Method.** We commence with a succinct introduction to gradient-based methods deployed in neural networks. Envision the training of a $L$-layer feedforward neural network, where each layer $l \in 1, 2, \ldots, L$ accepts $h_{l-1}$ as an input, generating an activation $h_l = F_l(h_{l-1}; w_l)$ with weight $w_l \in \mathbb{R}^{d_l}$. We represent all parameters by $w = [w_1^\intercal, w_2^\intercal, \ldots, w_L^\intercal]^\intercal \in \mathbb{R}^d$ and the output of the network by $h_L = F(h_0, w)$, where $h_0$ symbolizes the input data $x$. Given a loss function $f$ and targets $y$, the objective problem is as follows:

$$\min_w f(F(x; w), y) \tag{1}$$

Here, we use $f(x; w)$ in subsequent sections for the sake of brevity.

Gradient-based methods are widely used to optimize deep learning problems. At iteration $t$, we feed data sample $x_{i(t)}$ into the network, where $i(t)$ signifies the sample's index. As per the principles of stochastic gradient descent (SGD), we define the network parameters as follows:

$$w_l^{t+1} = w_l^t - \gamma_t \nabla f_{l,x_{i(t)}}(w^t), \quad \forall l \in \{1, 2, \ldots, L\} \tag{2}$$

Here, $\gamma_t \in \mathbb{R}$ is the stepsize and $\nabla f_{l,x_{i(t)}}(w^t) \in \mathbb{R}^{d_l}$ is the gradient of the loss function with respect to the weights at layer $l$ and data sample $x_{i(t)}$.

The crux of the matter is obtaining $\nabla f_{l,x_{i(t)}}(w^t)$. Both scientific and industrial sectors lean towards backpropagation underpinned by automatic differentiation (AD). However, FGD could be another alternative, because it can approximate the true gradient without bias by using forward-mode AD, and importantly has low memory consumption as it only preserve the intermediate variable passed from previous layer, while backpropagation need to store intermediate variable in each layer.

**Forward Gradient Descent.** Let $J_{F_l}$ represent the Jacobian matrix of layer $l$ while $u_{w_l}$ represent the random perturbation on $w_l$ (more precisely, tangent vector around $w_l$), we can calculate the JVP value $o_l$, in each layer sequentially by

$$h_l = F_l(h_{l-1}; w_l) \in \mathbb{R}^{d_{h_l}}, \tag{3}$$

$$o_l = J_{F_l}(h_{l-1}, w_l)u_l \in \mathbb{R}^{d_{h_l}}, \quad \text{where} \quad u_l = [o_{l-1}^\intercal, u_{w_l}^\intercal]^\intercal \in \mathbb{R}^{d_{h_{l-1}}+d_l} \tag{4}$$

Mathematically, $o_l$ is the directional derivative of $F_l$ along $u_l$; intuitively, $o_l$ can be interpreted as the influence of the perturbation $u_{w_l}$ on function value $h_l$ (we set $o_0$ to be 0 since we don't need to calculate the JVP value with respect to the input data). This process can be arranged within the forward pass such that (3), (4) are computed in the same pass. Also, since we don't need to propagate backward, $h_{l-1}$ and $o_{l-1}$ are thrown away right after the computation of $h_l$ and $o_l$. Then we can approximate the true gradient $\nabla f(x; w)$ unbiasedly (Lemma 3.1) by $\left(\frac{\partial f}{\partial h_L} o_L\right) u$, where $u = [u_{w_1}^\intercal, \ldots, u_{w_L}^\intercal]^\intercal$.

**Lemma 3.1.** *Let $u_{w_l} \in \mathbb{R}^{d_l}, l \in \{1, 2, \ldots, L\}$ be a normally distributed random Gaussian vectors, then the forward gradient computed through Eq.(3), (4) is an unbiased estimate of the true gradient*

$$\nabla f(x; w) = \mathbb{E}_u \left[ \left( \frac{\partial f}{\partial h_L} o_L \right) u \right].$$

More specifically, each layer $l$ receives $o_L$ from the last layer and updates its own parameters locally by using $\mathbb{E}_{u_{w_l}}(u_{w_l} \cdot o_L)$. We can then rewrite (2) as :

$$w_l^{t+1} = w_l^t - \gamma_t \left( \left( \frac{\partial f}{\partial h_L} o_L^t \right) u_{w_l}^t \right) \tag{5}$$

**Forward Locking.** It is evident from Eq.(4) that the computation in layer $l$ depends on the activation and JVP value $h_{l-1}, o_{l-1}$ from layer $l-1$. This dependency creates a "lock" that prevents all layers from updating before receiving the output from dependent layers, thus serializing the computation in the forward pass (refer to Figure 1 for illustration).

# 4  AsyncFGD

In this section, we propose an innovative approach that utilizes module-wise staleness to untether the dependencies inherent in FGD. This method, which we've named AsyncFGD, facilitates the simultaneous execution of tasks originating from disparate iterations. Suppose a $L$-layer feedforward neural network is divided into $K$ modules, with each module comprising a set of consecutive layers and their respective parameters. Consequently, we have a configuration such that $w = [w_{\mathcal{G}(0)}^\intercal, w_{\mathcal{G}(1)}^\intercal, \cdots, w_{\mathcal{G}(K-1)}^\intercal]^\intercal$, with $\mathcal{G}(k)$ denoting the layer indices in group $k$.

Now, let's delve into the details of how AsyncFGD untethers iteration dependencies and expedites the training process.

## 4.1  Detaching Iteration Dependency

**Forward.** At the timestamp $t$, the data sample $x_i(t)$ is pumped to the network. In contrast to sequential FGD [1], AsyncFGD permits modules to concurrently compute tasks, each originally belonging to a distinct iteration. All modules, with the exception of the last, operate using delayed parameters. We designate the activation and its JVP value originally ascribed to iteration $t$ in Eq.(3), (4) as $\hat{o}_l^t, \hat{h}_l^t$ respectively. Though the superscript $t$ is no longer time-dependent, it maintains its role in indicating sequential order. Consider $L_k \in \mathcal{G}(k)$ as the final layer in module $k$ and $f_k$ as the activation of this last layer. The computation in module $k \in 0, 1, \cdots, K-1$ at timestamp $t$ is defined recursively as follows:

$$\hat{h}_{L_k}^{t-k} = f_k \left( \hat{h}_{L_{k-1}}^{t-k}; w_{\mathcal{G}(k)}^{t-2K+k+2} \right) \tag{6}$$

$$\hat{o}_{L_k}^{t-k} = J_{f_k}\left(\hat{h}_{L_{k-1}}^{t-k}; w_{\mathcal{G}(k)}^{t-2K+k+2}\right) u_{\mathcal{G}(k)}^{t-k}, \quad \text{where } u_{\mathcal{G}(k)}^{t-k} = \left[\hat{o}_{L_{k-1}}^{t-k\mathsf{T}}, u_{w_{\mathcal{G}(k)}}^{t-k\mathsf{T}}\right]^{\mathsf{T}}. \tag{7}$$

Concurrently, each module receives and stores output from its dependent module for future computations.

**Update.** The update phase in AsyncFGD parallels that in FGD [1]. The final module broadcasts its JVP value, triggering each module to perform local updates to their parameters. To summarize, at timestamp $t$, we execute the following update:

$$w^{t-K+2} = w^{t-K+1} - \gamma_{t-K+1}\left(\hat{o}_{L_{K-1}}^{t-K+1} u_w^{t-K+1}\right) \tag{8}$$

**Staleness.** We measure the time delay in Eq.(6), (7) with $g(k) = K - 1 - k$, and $g(K-1) = 0$ indicates that last module employs up-to-date parameters.

This approach effectively disrupts the "lock-in" characteristic of FGD, facilitating a parallel forward pass. A comparative illustration of the execution details in sequential FGD and AsyncFGD is provided in Figure 1.

### 4.2 Stochastic AsyncFGD Alogrithm

To better illustrate the working state of AsyncFGD, we make the following definitions:

$$w^{t-K+1} := \begin{cases} w^0, & t-K+1 < 0 \\ w^{t-K+1}, & \text{otherwise} \end{cases}; \quad \text{and } \hat{h}_{L_k}^{t-k}, \hat{o}_{L_k}^{t-k} := \begin{cases} 0, 0, & t-k < 0 \\ \hat{h}_{L_k}^{t-k}, \hat{o}_{L_k}^{t-k}, & \text{otherwise} \end{cases} \tag{9}$$

Unlike FGD, AsyncFGD forwards the JVP value and activations by parameters in different time delays, which can be concluded as $f\left(x_{i(t-2K+2)}; w_{\mathcal{G}(0)}^{t-2K+2}; w_{\mathcal{G}(1)}^{t-2K+3}; \cdots; w_{\mathcal{G}(K-1)}^{t-K+1}\right)$. A detailed illustration of the AsyncFGD with $K = 3$ is shown in Appendix G.2. We summarize the proposed algorithm in Algorithm 1 by example of sampling one tangent vector per iteration.

---

**Algorithm 1** AsyncFGD-SGD

---

**Initialize:** Stepsize sequence $\{\gamma_t\}_{t=K-1}^{T-1}$, weight $w^0 = \left[w_{\mathcal{G}(0)}^0, \cdots, w_{\mathcal{G}(K-1)}^0\right] \in \mathbb{R}^d$

1: **for** $t = 0, 1, \cdots, T-1$ **do**
2:     **for** $k = 0, 1, \cdots, K-1$ **in parallel do**
3:         Read $\hat{h}_{L_{k-1}}^{t-k}, \hat{o}_{L_{k-1}}^{t-k}$ from storage if $k \neq 0$
4:         Compute $\hat{h}_{L_k}^{t-k}, \hat{o}_{L_k}^{t-k}$
5:         Send $\hat{h}_{L_k}^{t-k}, \hat{o}_{L_k}^{t-k}$ to next worker's storage if $k \neq K-1$
6:     **end for**
7:     Broadcast $\hat{o}_{L_{K-1}}^{t-K+1}$
8:     **for** $k = 0, 1, \cdots, K-1$ **in parallel do**
9:         Compute $\Delta w_{\mathcal{G}(k)}^{t-K+1} = \hat{o}_{L_{K-1}}^{t-K+1} u_{w_{\mathcal{G}(k)}}^{t-K+1}$
10:        Update $w_{\mathcal{G}(k)}^{t-K+2} = w_{\mathcal{G}(k)}^{t-K+1} - \gamma_{t-K+1}\Delta w_{\mathcal{G}(k)}^{t-K+1}$
11:     **end for**
12: **end for**

---

Additionally, the procedure in line 5 could overlap with the operations in line 7,9 and 10. We can also apply this approximation algorithm to gradient-based methods like Adam [18] with little modification to the original. Details can be found in Appendix G.1.

**Tangent checkpoint.** However, some workers, especially those which are closer to the input, store duplicated tangent vectors. To tackle this issue, we use tangent checkpoint,i.e., storing the seed of tangent vectors and reproducing them in the update stage.

**Integration with Efficient Transfer Learning** Although AsyncFGD offers several advantages over backpropagation, it shares a common challenge with random Zeroth-Order Optimization and Evolution Strategy methods: the variance of the approximation increases with the dimension of random perturbations[25]. Reducing the learning rate can help but may result in slower convergence. However, we observe that deploying models on edge devices typically involves fine-tuning rather

than training from scratch. Our method can flexibly incorporate the principles of efficient transfer learning by introducing a scaling factor $\alpha \in [0, 1]$ to the randomly sampled tangent:

$$u'_{w_l} = u_{w_l} \cdot \alpha_{w_l}$$

The modified $u'_{w_l}$ still supports an approximation of the true gradient, with the expectation of the modified estimator being $\alpha^2_{w_l} \nabla f(x; w_l)$. When $\alpha_{w_l}$ is set to 0, the corresponding parameter is "frozen," resulting in no perturbation or updates and a transparent transmission of JVP values. By adjusting $\alpha$ to various values, we can either fix or reduce the influence and learning of specific layers, aligning our approach with the objectives of efficient transfer learning.

### 4.3 Acceleration of AsyncFGD

When $K = 1$ the AsyncFGD is reduced to vanilla FGD without any time delay in parameters. When $K \geq 2$, we can distribute the network across multiple workers. Figure 2 shows the computational time of different algorithms in ideal conditions (i.e. the network is evenly distributed and the communication is overlapped by computation and update). $T_F$ denotes the time for forward pass and $T_U$ denotes the time for updates. It is easy to see that AsyncFGD can fully utilize the computation resources, thus achieving considerable speedup.

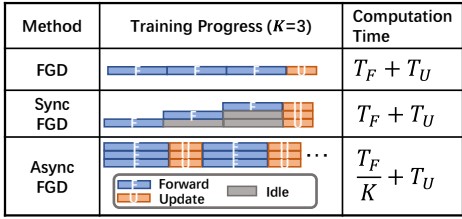

Figure 2: Comparison of computation time and training process when the network is deployed across $K$ workers. Since $T_F$ is much larger than $T_U$, AsyncFGD can achieve considerable speedup.

## 5 Convergence Analysis

In this section, we provide the convergence guarantee of Algorithm 1 to critical points in a non-convex setup. We first make the following basic assumptions for nonconvex optimization:

**Assumption 5.1.** The gradient of $f(w)$ is Lipschitz continuous with Lipschitz constant $L > 0$, *i.e.*,

$$\forall x, y \in \mathbb{R}^d, \|\nabla f(x) - \nabla f(y)\| \leq L\|x - y\|.$$

**Assumption 5.2.** The second moment of stochastic gradient is bounded, i.e., there exist a constant $M \geq 0$, for any sample $x_i$ and $\forall w \in \mathbb{R}^d$:

$$\|\nabla f_{x_i}(w)\|^2 \leq M.$$

**Lemma 5.3 (Mean and Variance).** *Let* $t' = t - K + 1$ *and diagonal matrices* $\mathbf{I}_0, \cdots, \mathbf{I}_k, \cdots, \mathbf{I}_{K-1} \in \mathbb{R}^{d \times d}$ *such that all the principle diagonal elements of* $\mathbf{I}_k$ *in* $\mathcal{G}(k)$ *are 1, and all the principle diagonal elements of* $I_k$ *in other than* $\mathcal{G}(k)$ *are 0. Then we can obtain the mean value and the variance of the forward gradient as follows,*

$$\mathbb{E}_{u^{t'}_{w_{\mathcal{G}(k)}}} \left( o^{t'}_{L_{K-1}} \cdot u^{t'}_{w_{\mathcal{G}(k)}} \right) = \nabla f_{\mathcal{G}(k), x_{i(t')}} \left( w^{t'-K+k+1} \right), \tag{10}$$

$$\mathbb{E}_{u^{t'}_w} \left\| \sum_{k=0}^{K-1} \mathbf{I}_k \cdot \hat{o}^{t'}_{L_{K-1}} \cdot u^{t'}_{w_{\mathcal{G}(k)}} \right\|^2 \leq (d+4) \left\| \sum_{k=0}^{K-1} \mathbf{I}_k \nabla f_{\mathcal{G}(k), x_{i(t')}} \left( w^{t'-K+k+1} \right) \right\|^2. \tag{11}$$

*Remark* 5.4. Note that, from modules 0 to $K-1$, the corresponding time delays are from $K-1$ to 0. Specifically, in timestamp $t'$, when $k = K-1$, $\mathbb{E}_{u^{t'}_{w_{\mathcal{G}(K-1)}}} \left( o^{t'}_L \cdot u^{t'}_{w_{\mathcal{G}(K-1)}} \right) = \nabla f_{\mathcal{G}(K-1), x_{i(t')}} \left( w^{t'} \right)$ indicates that the forward gradient in module $K-1$ is an unbiased estimation of the up-to-date gradient.

Under Assumption 5.1 and 5.2, we obtain the following descent lemma about the objective function value:

**Lemma 5.5.** *Assume that Assumption 5.1 and 5.2 hold. In addition, let* $t' = t - K + 1, \sigma := \max_{t'} \frac{\gamma_{\max\{0, t'-K+1\}}}{\gamma_{t'}}, M_K = KM + \sigma K^4 M$ *and choose* $\gamma_{t'} \leq \frac{1}{L}$. *The iterations in Algorithm 1 satisfy the following descent property in expectation,* $\forall t' \in \mathbb{N}$:

$$\mathbb{E} \left[ f(w^{t'+1}) \right] - f(w^{t'}) \leq -\frac{\gamma_{t'}}{2} \|\nabla f(w^{t'})\|^2 + 4(d+4)L\gamma^2_{t'}M_K, \tag{12}$$

**Theorem 5.6.** *Assume Assumption 5.1 and 5.2 hold and the fixed stepsize sequence $\{\gamma_{t'}\}$ satisfies $\gamma_{t'} = \gamma \leq \frac{1}{L}, \forall t' \in \{0, 1, \ldots, T-1\}$. In addition, we assume $w^*$ to be the optimal solution to $f(w)$ and let $t' = t - K + 1$, $\sigma = 1$ such that $M_K = KM + K^4 M$. Then, the output of Algorithm 1 satisfies that:*

$$\frac{1}{T} \sum_{t'=0}^{T-1} \mathbb{E}\|\nabla f(w^{t'})\|^2 \leq \frac{2(f(w^0) - f(w^*))}{\gamma T} + 4(d+4)L\gamma M_K. \tag{13}$$

**Theorem 5.7.** *Assume Assumption 5.1 and 5.2 hold and the diminishing stepsize sequence $\{\gamma_{t'}\}$ satisfies $\gamma_{t'} = \frac{\gamma_0}{t'+1} \leq \frac{1}{L}$. In addition, we assume $w^*$ to be the optimal solution to $f(w)$ and let $t' = t - K + 1$, $\sigma = K$ such that $M_K = KM + K^5 M$. Let $\Gamma_T = \sum_{t'=0}^{T-1} \gamma_{t'}$, the output of Algorithm 1 satisfies that:*

$$\frac{1}{\Gamma_T} \sum_{t'=0}^{T-1} \gamma_{t'} \mathbb{E}\|\nabla f(w^{t'})\|^2 \leq \frac{2\left(f(w^0) - f(w^*)\right)}{\Gamma_T} + \frac{\sum_{t'=0}^{T-1} 4(d+4)\gamma_{t'}^2 L M_K}{\Gamma_T} \tag{14}$$

*Remark* 5.8. Since the stepsize sequence $\gamma_t = \frac{\gamma_0}{t+1}$ satisfies that $\lim_{T \to \infty} \sum_{t=0}^{T-1} \gamma_t = \infty$, $\lim_{T \to \infty} \sum_{t=0}^{T-1} \gamma_t^2 < \infty$, when $T \to \infty$, the RHS of Eq.(14) will converges to 0. Let $w^s$ be randomly chosen from $\{w^{t'}\}_{t'=0}^{T-1}$ with probabilities proportional to $\{\gamma_{t'}\}_{t'=0}^{T-1}$, then we have $\lim_{s \to \infty} \mathbb{E}\|\nabla f(w^s)\|^2 = 0$.

## 6  Experiments

This section embarks on a meticulous examination of our proposed method, AsyncFGD. We assess its performance through three distinct facets: memory consumption, acceleration rate, and accuracy. We initiate our analysis by outlining our experimental setup. To validate the efficacy of applying directional derivatives and utilizing module-wise stale parameters, we contrast AsyncFGD with an array of alternative methods encompassing backpropagation, conventional FGD, and other backpropagation-free algorithms. Subsequently, our focus shifts towards scrutinizing the potential of AsyncFGD within the sphere of efficient transfer learning, conducting experiments on prevalent efficient networks. Memory consumption represents another cardinal aspect that we explore, benchmarking AsyncFGD against popular architectures and unit layers like fully-connected layers, RNN cells, and convolutional layers. Concluding our empirical investigation, we assess the speedup of our method relative to other parallel strategies under a diverse set of conditions across multiple platforms.

### 6.1  Experimental Setup

**Methods.** We contrast our proposal's memory footprint with Backpropagation, Sublinear [3], Backpropagation through time, and Memory Efficient BPTT [7]. Accuracy-wise, we compare with backpropagation-free methods: Feedback Alignment (FA) [27], Direct Feedback Alignment (DFA) [26], Direct Random Tangent Propagation, and Error-sign-based Direct Feedback Alignment (sDFA) [6]. We also apply parallelization to FGD through FGD-DP (data parallelism) and FGD-MP (model parallelism).

**Platform.** Experiments utilize Python 3.8 and Pytorch, primarily on nVidia's AGX Orin. Additional results on alternate platforms are in the appendix.

**Training.** Batch size is 64 unless noted. The optimal learning rate (chosen from $\{1e-5, 1e-4, 1e-3, 1e-2\}$ with Adam optimizer [18]) is based on validation performance. The parameter $\alpha$ is initially set to 1 for the classifier, with others at 0 for the first 10 epochs. Subsequently, $\alpha$ is gradually increased to 0.15 for specific layers. More details are in the appendix.

### 6.2  Effectiveness of Directional Derivative and Asynchronism

This section documents our experimentation on the effectiveness of using random directional derivative to approximate the true gradient by contrasting it with other BP-free algorithms. Furthermore, we demonstrate that the consistency remains intact when using module-wise stale parameters to uncouple the dependencies, comparing AsyncFGD with vanilla FGD. Results in Table 1 indicate that AsyncFGD can produce results closely aligned with vanilla FGD. Notably, FGD and AsyncFGD yield

Table 1: Comparison for AsyncFGD with other BP-free methods. ConvS and FCS refers to small convlutional network and full-connected network while ConvL and FCL refers to their slightly bigger couterpart. Different activation functions are marked as subscript. Details of network architecutre can be found in Appendix H.2

| Dataset | Model | BP | BP-free | | | | | |
| --- | --- | --- | --- | --- | --- | --- | --- | --- |
| | | | FA | DFA | sDFA | DRTP | FGD | Async |
| MNIST | ConvS$_{Tanh}$ | 98.7 | 88.1 | 95.9 | **96.8** | 95.4 | 94.6 | 94.4 |
| | ConvS$_{ReLU}$ | 99.2 | 12.0 | 11.5 | 13.8 | 13.6 | **95.5** | **95.5** |
| | ConvL$_{Tanh}$ | 99.3 | 8.7 | 92.2 | 93.4 | 92.6 | **94.4** | 94.2 |
| | ConvL$_{ReLU}$ | 99.3 | 89.7 | 93.0 | 93.1 | **93.2** | 93.0 | **93.2** |
| | FCS$_{Tanh}$ | 98.9 | 83.2 | **95.6** | 94.2 | 94.5 | 94.4 | 94.3 |
| | FCSl$_{ReLU}$ | 98.5 | 8.8 | 10.0 | 10.8 | 9.9 | 93.6 | **93.7** |
| | FCL$_{Tanh}$ | 98.8 | 89.8 | 93.0 | 92.0 | 92.4 | **95.2** | 95.4 |
| | FCL$_{ReLU}$ | 99.3 | 86.3 | 93.8 | 94.3 | 94.1 | 94.8 | **95.3** |
| CIFAR-10 | ConvS$_{Tanh}$ | 69.1 | 33.4 | 56.5 | 57.4 | **57.6** | 46.5 | 46.0 |
| | ConvS$_{ReLU}$ | 69.3 | 12.0 | 11.5 | 10.8 | 12.0 | **40.0** | 39.7 |
| | ConvL$_{Tanh}$ | 71.0 | 40.4 | 42.0 | 43.6 | 44.1 | **47.3** | 47.3 |
| | ConvL$_{ReLU}$ | 71.2 | 40.4 | 42.0 | 43.6 | 44.1 | **44.2** | 44.1 |
| | FCS$_{Tanh}$ | **47.8** | 46.2 | 46.4 | 46.0 | 46.2 | 42.0 | 42.3 |
| | FCS$_{ReLU}$ | 52.7 | 10.2 | 12.0 | 10.0 | 10.3 | **43.7** | 43.6 |
| | FCL$_{Tanh}$ | 54.4 | 17.4 | 44.0 | 44.3 | 45.5 | **47.2** | **47.2** |
| | FCL$_{ReLU}$ | 55.3 | 40.4 | 42.0 | 43.6 | 44.1 | 46.0 | **46.7** |

the most optimal outcomes when the network is more complex, or when we employ ReLU as the activation function devoid of batchnorm layers (results from ConvS$_{ReLU}$ and FCS$_{ReLU}$), situations where FA, DFA and sDFA often fail to propagate an effective error message. Backpropagation results are also furnished solely as a reference to true gradient results. However, when the network grows larger, all BP-free algorithms grapple with variance. The use of larger networks results in only minor improvements compared to BP. We try to address this challenge through efficient transfer learning in the subsequent section.

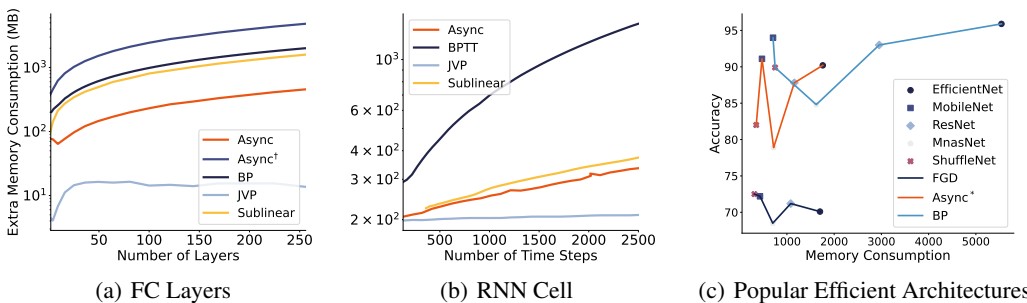

(a) FC Layers     (b) RNN Cell     (c) Popular Efficient Architectures

Figure 3: Memory footprint comparison across methods. Async$^{\dagger}$ is AsyncFGD without tangent checkpoint while Async$^{*}$ refers to AsyncFGD using efficient training strategy. (a) FC layer memory vs. layer count; (b) RNN memory vs. sequential length; (c) Accuracy vs. memory on efficient architectures.

## 6.3 Efficacy of Efficient Transfer Learning

In this segment, we delve into the efficacy of amalgamating AsyncFGD with efficient transfer learning, focusing on popular architectures like ResNet-18(Res-18) [9]; MoblieNet (Mobile)[11]; MnasNet(Mnas)[36] and ShuffleNet(Shuffle)[45] with their lightweight counterpart. The models are fine-tuned with weights pre-trained on Imagenet. AsyncFGD$^{\dagger}$ denotes AsyncFGD utilizing the efficient training strategy. As can be observed from Table 2, the application of an efficient transfer learning strategy brings about substantial performance enhancement, yielding superior results

compared to training with perturbation on the full model. Ablation studies on $\alpha$ provided in Appendix I.1 also shows that, compared to optimizing subset of the model, FGD suffers more from variance.

## 6.4 Memory Footprint

As illustrated in Fig 3(c), when we plot accuracy against memory consumption, it is evident that AsyncFGD employs approximately one-third of the memory while maintaining close accuracy. Further experimentation on memory consumption with respect to the computing unit reveals, as shown in Fig 3(a) and Fig 3(b), that the additional memory consumption in AsyncFGD mainly serves as placeholders for random tangents, while the JVP computation consumes a negligible amount of additional memory. Memory consumption of other basic units like CNN and batch norm layers, are provided in Appendix.

## 6.5 Acceleration on Input Stream

In this final section, we assess the acceleration of ResNet-18 with varying $K$. In this setting, the batch size is set to 4 to better reflect the mechanism of streamlined input on edge device. As demonstrated in 4, AsyncFGD, by detaching dependencies in the forward pass, can outperform other parallel strategies in terms of acceleration rate. While pipeline parallelism is fast, the locking within the forward pass still induces overhead for synchronization, ultimately leading to lower hardware utilization and speed. Results pertaining to different network architectures and other platforms like CPU and GPU as well as more generalized case for larger batch size can be found in the Appendix I.2.

Table 2: Results for different algorithms in transfer learning. Async* refers to using efficient transfer learning strategy.

| Dataset | Model | BP | FGD | Async | Async* |
|---------|-------|------|------|-------|--------|
| MNIST | Res-18 | 98.5 | 90.6 | 90.4 | **96.4** |
| | Mobile | 99.2 | 89.3 | 88.4 | **97.1** |
| | Efficient | 99.2 | 90.4 | 90.1 | **95.9** |
| | Mnas | 98.9 | 86.6 | 86.3 | **96.0** |
| | Shuffle | 99.0 | 85.8 | 85.8 | **96.4** |
| CIFAR | Res-18 | 93.0 | 71.2 | 71.2 | **87.8** |
| | Mobile | 94.0 | 72.3 | 72.2 | **91.1** |
| | Efficient | 94.9 | 70.2 | 70.1 | **90.2** |
| | Mnas | 84.2 | 68.8 | 68.5 | **78.9** |
| | Shuffle | 89.9 | 72.5 | 72.5 | **82.0** |
| FMNIST | Res-18 | 94.2 | 80.2 | 80.2 | **88.0** |
| | Mobile | 93.2 | 82.3 | 83.1 | **90.6** |
| | Efficient | 92.8 | 79.8 | 79.8 | **90.4** |
| | Mnas | 92.1 | 77.1 | 77.0 | **87.0** |
| | Shuffle | 92.8 | 78.4 | 78.4 | **87.3** |

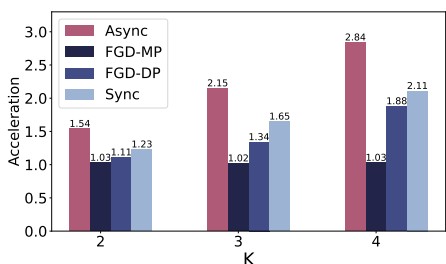

(a) Cluster with 4 GPUs

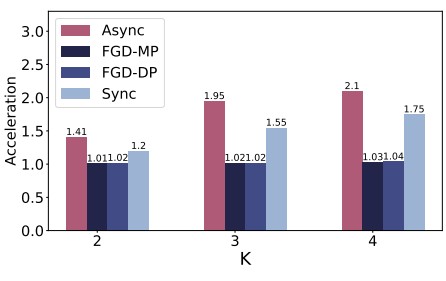

(b) Single Embedded Device

Figure 4: Comparison for acceleration of different parallel methods.

## 7 Limitations and Discussion

While AsyncFGD offers computational benefits, it is not without limitations. A key drawback is its inferior performance compared to traditional Backpropagation (BP). This performance disparity is largely attributed to the use of randomly sampled directional derivatives in the forward gradient computation, aligning AsyncFGD with Zeroth-Order (ZO) optimization methods and evolutionary strategies. This sampling introduces a significant source of gradient variance, a challenge that is part of a larger problem in the field of stochastic optimization. However, we take encouragement from

recent advancements aimed at reducing this variance, some of which have even facilitated the training of large-scale models [31].

Another constraint pertains to the availability of idle workers on edge devices—a condition that is not universally applicable given the wide variety of edge computing environments. These can span from IoT chips with limited computational resources, where even deploying a standard deep learning model can be problematic, to high-capacity micro-computers used in autonomous vehicles.

Nevertheless, our experimental findings suggest that AsyncFGD is particularly beneficial for specific edge computing scenarios. In such settings, it may serve as a viable alternative for reducing memory usage while fully leveraging available computational resources.

## 8   Conclusion

In the present paper, we introduce AsyncFGD, an innovative approach designed to shatter the shackles of locking within the forward pass in FGD. This is achieved by incorporating module-wise stale parameters, simultaneously retaining the advantage of minimized memory consumption. In the theoretical segment, we offer a lucid analysis of this partially ordered staleness, demonstrating that our proposed method is capable of converging to critical points even in the face of non-convex problems. We further extend our algorithm to efficient transfer learning by introducing a scale parameter. Our experimental reveals that a sublinear acceleration can be accomplished, without compromising accuracy as well as huge performance gain when utilizing efficient transfer learning strategy. While the exploration of large models employing extensive datasets will undoubtedly continue to rely on backpropagation [10], we assert that the potential of asynchronous algorithms predicated on forward computation should not be overlooked. It offers a promising avenue for fully harnessing limited resources in on-device learning scenarios.

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

# A Appendix

The organization of the Appendix is as follows: Appendix sections B to F provide detailed proofs of the lemmas and theorems presented in the main text. This is followed by additional insights, including operational specifics of working with Adam and details on how AsyncFGD can be extended for parallel processing on recursive networks, covered in Appendix G. Lastly, Appendix H and I provide a comprehensive view of the training process and additional experimental results, respectively.

# B Proof of Lemma 3.1

According to the update rule of Eq.(3), (4), we have

$$
\begin{aligned}
h_1 =& F_1(h_0, w_1) = F_1(x, w_1) \\
o_1 =& J_{F_1}(h_0, w_1)[o_0^\mathsf{T}, u_{w_1}^\mathsf{T}]^\mathsf{T} = J_{F_1}(h_0)o_0 + J_{F_1}(w_1)u_{w_1} = J_{F_1}(w_1)u_{w_1} \\
h_2 =& F_2(h_1, w_2) \\
o_2 =& J_{F_2}(h_1, w_2)[o_1^\mathsf{T}, u_{w_2}^\mathsf{T}]^\mathsf{T} = J_{F_2}(h_1)o_1 + J_{F_2}(w_2)u_{w_2} = J_{F_2}(h_1)J_{F_1}(w_1)u_{w_1} + J_{F_2}(w_2)u_{w_2} \\
=& J_{F_2}(w_1)u_{w_1} + J_{F_2}(w_2)u_{w_2} \\
&\cdots
\end{aligned}
$$

$$
o_L = \sum_{l=1}^{L} J_{F_L}(w_l)u_{w_l}
$$

Then for any $l \in \{1, 2, \ldots, L\}$, take expectation with respect to $u_{w_l}$, we have

$$
\mathbb{E}_{u_{w_l}}(o_L \cdot u_{w_l}) = \mathbb{E}_{u_{w_l}} \left[ (J_{F_L}(w_l)u_{w_l}) \cdot u_{w_l} + \sum_{k \neq l} (J_{F_L}(w_k)u_{w_k}) \cdot u_{w_l} \right]
$$

Note that

$$
\begin{aligned}
\mathbb{E}_{u_{w_l}} \left[ (J_{F_L}(w_l)u_{w_l}) \cdot u_{w_l} \right] =& \mathbb{E}_{u_{w_l}} \left( \begin{bmatrix} \frac{\partial J_{F_L,1}}{\partial w_{l,1}} & \frac{\partial J_{F_L,1}}{\partial w_{l,2}} & \cdots & \frac{\partial J_{F_L,1}}{\partial w_{l,d_l}} \\ \frac{\partial J_{F_L,2}}{\partial w_{l,1}} & \frac{\partial J_{F_L,2}}{\partial w_{l,2}} & \cdots & \frac{\partial J_{F_L,2}}{\partial w_{l,d_l}} \\ \vdots & \vdots & \ddots & \vdots \\ \frac{\partial J_{F_L,d_{h_L}}}{\partial w_{l,1}} & \frac{\partial J_{F_L,d_{h_L}}}{\partial w_{l,2}} & \cdots & \frac{\partial J_{F_L,d_{h_L}}}{\partial w_{l,d_l}} \end{bmatrix} \begin{bmatrix} u_{w_l,1} \\ u_{w_l,2} \\ \vdots \\ u_{w_l,d_l} \end{bmatrix} \cdot \begin{bmatrix} u_{w_l,1} & u_{w_l,2} & \cdots u_{w_l,d_l} \end{bmatrix} \right) \\
=& \mathbb{E}_{u_{w_l}} \left( \begin{bmatrix} \sum_{i=1}^{d_l} \frac{\partial J_{F_L,1}}{\partial w_{l,i}} \cdot u_{w_l,i} \\ \sum_{i=1}^{d_l} \frac{\partial J_{F_L,2}}{\partial w_{l,i}} \cdot u_{w_l,i} \\ \vdots \\ \sum_{i=1}^{d_l} \frac{\partial J_{F_L,d_{h_L}}}{\partial w_{l,i}} \cdot u_{w_l,i} \end{bmatrix} \cdot \begin{bmatrix} u_{w_l,1} & u_{w_l,2} & \cdots u_{w_l,d_l} \end{bmatrix} \right) \\
=& \mathbb{E}_{u_{w_l}}(D),
\end{aligned}
$$

where

$$
D_{m,n} = \left( \sum_{i=1}^{d_l} \frac{\partial J_{F_L,m}}{\partial w_{l,i}} \cdot u_{w_l,i} \right) u_{w_l,n} = \frac{\partial J_{F_L,m}}{\partial w_{l,n}} u_{w_l,n}^2 + \sum_{k \neq n} \frac{\partial J_{F_L,m}}{\partial w_{l,k}} u_{w_l,k} u_{w_l,n}
$$

with $m \in \{1, 2, \ldots, d_{h_L}\}, n \in \{1, 2, \ldots, d_l\}$. Since each $u_{w_l} \sim \mathbb{N}(0, I)$, we have

$$
\mathbb{E}_{u_{w_l}}(D_{m,n}) = \frac{\partial J_{F_L,m}}{\partial w_{l,n}}, \quad \mathbb{E}_{u_{w_l}}(D) = J_{F_L}(w_l).
$$

Similarly, we can prove that $\mathbb{E}_{u_{w_l}} \left[ \sum_{k \neq l} (J_{F_L}(w_k)u_{w_k}) \cdot u_{w_l} \right] = 0 \in \mathbb{R}^{d_{h_L} \times d_l}$. So we have,

$$
\mathbb{E} \left( \frac{\partial f}{\partial F_L} \cdot o_L \cdot u_{w_l} \right) = \frac{\partial f}{\partial F_L} J_{F_L}(w_l) = \nabla_{l,x} f(w) \in \mathbb{R}^{d_l}
$$

# C   Proof of Lemma 5.3

***Proof of Mean.***

$$\hat{h}_1^{t-K+1} = F_1(h_0^{t-K+1}, w_1^{t-2K+2}) = F_1(x_{i(t-K+1)}, w_1^{t-2K+2})$$

$$\hat{o}_1^{t-K+1} = J_{F_1}(h_0^{t-K+1}, w_1^{t-2K+2})[o_0^{\mathsf{T}}, u_{w_1}^{t-K+1\mathsf{T}}]^{\mathsf{T}} = J_{F_1}(h_0^{t-K+1})o_0 + J_{F_1}(w_1^{t-2K+2})u_{w_1}^{t-K+1}$$

$$= J_{F_1}(w_1^{t-2K+2})u_{w_1}^{t-K+1}$$

$$\hat{h}_2^{t-K+1} = F_2(\hat{h}_1^{t-K+1}, w_2^{t-2K+2})$$

$$\hat{o}_2^{t-K+1} = J_{F_2}(h_1^{t-K+1}, w_2^{t-K+1})[o_1^{t-K+1\mathsf{T}}, u_{w_2}^{t-K+1\mathsf{T}}]^{\mathsf{T}} = J_{F_2}(h_1^{t-K+1})o_1^{t-K+1} + J_{F_2}(w_2^{t-2K+2})u_{w_2}^{t-K+1}$$

$$= J_{F_2}(h_1^{t-K+1})J_{F_1}(w_1^{t-2K+2})u_{w_1}^{t-K+1} + J_{F_2}(w_2)u_{w_2^{t-2K+2}}^{t-K+1}$$

$$\cdots\cdots$$

$$\hat{o}_{L_0}^{t-K+1} = \sum_{l=1}^{L_0} J_{f_0}(w_l^{t-2K+2})u_{w_l}^{t-K+1} = \sum_{l\in\mathcal{G}(0)} J_{f_0}(w_l^{t-2K+2})u_{w_l}^{t-K+1}$$

$$\cdots\cdots$$

$$\hat{o}_{L_t}^{t-K+1} = \sum_{l\in\mathcal{G}(0)} J_{f_t}(w_l^{t-2K+2})u_{w_l}^{t-K+1} + \sum_{l\in\mathcal{G}(1)} J_{f_t}(w_l^{t-2K+3})u_{w_l}^{t-K+1} + \cdots + \sum_{l\in\mathcal{G}(t)} J_{f_t}(w_l^{t-2K+t+2})u_{w_l}^{t-K+1}$$

$$= \sum_{k=0}^{t}\sum_{l\in\mathcal{G}(k)} J_{f_t}(w_l^{t-2K+k+2})u_{w_l}^{t-K+1}$$

$$\cdots\cdots$$

$$\hat{o}_{L_{K-1}}^{t-K+1} = \sum_{k=0}^{K-1}\sum_{l\in\mathcal{G}(k)} J_{f_{K-1}}(w_l^{t-2K+k+2}) = \sum_{k=0}^{K-1}\sum_{l\in\mathcal{G}(k)} J_f(w_l^{t-2K+k+2})u_{w_l}^{t-K+1}$$

$$= \sum_{k=0}^{K-1}\sum_{l\in\mathcal{G}(k)} \nabla f_{l,x_{i(t-K+1)}}(w^{t-2K+k+2})^{\mathsf{T}}u_{w_l}^{t-K+1}$$

Take expectation with respect to $u_{w_l}^{t-K+1}$, where $l \in \mathcal{G}(k)$, we have

$$\mathbb{E}_{u_{w_l}^{t-K+1}}(\hat{o}_{L_{K-1}}^{t-K+1} \cdot u_{w_l}^{t-K+1}) = \nabla f_{l,x_{i(t-K+1)}}(w^{t-2K+k+2})$$

So we have,

$$\mathbb{E}_{u_{w_{\mathcal{G}(k)}}^{t-K+1}}(\hat{o}_{L_{K-1}}^{t-K+1} \cdot u_{w_l}^{t-K+1}) = \nabla f_{\mathcal{G}(k),x_{i(t-K+1)}}(w^{t-2K+k+2})$$

$\square$

**Lemma C.1** ([25], Theorem 3). *Let $g_u(x) = \langle \nabla f(x), u \rangle u$, where $u \in \mathbb{R}^d$ is a normally distributed Gaussian vector, then we have*

$$\mathbb{E}_u\|g_u(x)\|^2 \le (d+4)\|\nabla f(x)\|^2$$

**Lemma C.2.** *Let $g_{u_1,u_2}(x) = \langle \nabla f(x), u_1 \rangle u_2$, where $u_1 \in \mathbb{R}^{d_1}, u_2 \in \mathbb{R}^{d_2}$ are two **i.i.d.** normally distributed Gaussian vectors, then we have*

$$\mathbb{E}_{u_1,u_2}\|g_{u_1,u_2}(x)\|^2 \le d_2\|\nabla f(x)\|^2$$

*Proof.*

$$\mathbb{E}_{u_1,u_2}\|g_{u_1,u_2}(x)\|^2 = \mathbb{E}_{u_1,u_2}\|\langle \nabla f(x), u_1 \rangle u_2\|^2 = \mathbb{E}_{u_1,u_2}\left(\langle \nabla f(x), u_1 \rangle^2 \|u_2\|^2\right)$$

$$= \mathbb{E}_{u_1}\left(\langle \nabla f(x), u_1 \rangle^2\right) \cdot \mathbb{E}_{u_2}\left(\|u_2\|^2\right)$$

$$\le d_2\mathbb{E}_{u_1}\left(\langle \nabla f(x), u_1 \rangle^2\right) = d_2\mathbb{E}_{u_1}\left(\sum_{i=1}^{d_1} \nabla_i f(x)u_{1,i}\right)^2$$

$$=d_2\mathbb{E}_{u_1}\left(\sum_{i=1}^{d_1}\nabla_i^2 f(x)u_{1,i}^2+\sum_{i\neq j}\nabla_i f(x)\nabla_j f(x)u_{1,i}u_{1,j}\right)$$

$$=d_2\|\nabla f(x)\|^2$$

where the first inequality is due to Lemma 1 in [25].

$\square$

*Proof of Variance.*

$$\mathbb{E}_{u_w^{t'}}\left\|\sum_{k=0}^{K-1}\mathbf{I}_k\cdot\hat{o}_{L_{K-1}}^{t'}\cdot u_{w_{\mathcal{G}(k)}}^{t'}\right\|^2$$

$$=\mathbb{E}_{u_w^{t'}}\sum_{k=0}^{K-1}\left\|\hat{o}_{L_{K-1}}^{t'}\cdot u_{\mathcal{G}(k)}^{t'}\right\|^2=\mathbb{E}_{u_w^{t'}}\sum_{k=0}^{K-1}\left\|\left(\sum_{j=0}^{K-1}\left\langle\nabla f_{\mathcal{G}(j),x_{i(t')}}(w^{t'-K+j+1}),u_{w_{\mathcal{G}(j)}}^{t'}\right\rangle\right)\cdot u_{w_{\mathcal{G}(k)}}^{t'}\right\|^2$$

$$=\mathbb{E}_{u_w^{t'}}\sum_{k=0}^{K-1}\left[\left(\sum_{j=0}^{K-1}\left\langle\nabla f_{\mathcal{G}(j),x_{i(t')}}(w^{t'-K+j+1}),u_{w_{\mathcal{G}(j)}}^{t'}\right\rangle\right)^2\cdot\left\|u_{\mathcal{G}(k)}^{t'}\right\|^2\right]$$

$$=\mathbb{E}_{u_w^{t'}}\sum_{k=0}^{K-1}\left[\left(\left\langle\nabla f_{\mathcal{G}(k),x_{i(t')}}(w^{t'-K+k+1}),u_{w_{\mathcal{G}(k)}}^{t'}\right\rangle^2+\sum_{j\neq k}\left\langle\nabla f_{\mathcal{G}(j),x_{i(t')}}(w^{t'-K+j+1}),u_{w_{\mathcal{G}(j)}}^{t'}\right\rangle^2\right.\right.$$

$$\left.\left.+\sum_{m\neq n}\left\langle\nabla f_{\mathcal{G}(m),x_{i(t')}}(w^{t'-K+m+1}),u_{w_{\mathcal{G}(m)}}^{t'}\right\rangle\cdot\left\langle\nabla f_{\mathcal{G}(n),x_{i(t')}}(w^{t'-K+n+1}),u_{w_{\mathcal{G}(n)}}^{t'}\right\rangle\right)\cdot\left\|u_{\mathcal{G}(k)}^{t'}\right\|^2\right]$$

$$=\mathbb{E}_{u_w^{t'}}\sum_{k=0}^{K-1}\left[\left(\left\langle\nabla f_{\mathcal{G}(k),x_{i(t')}}(w^{t'-K+k+1}),u_{w_{\mathcal{G}(k)}}^{t'}\right\rangle^2+\sum_{j\neq k}\left\langle\nabla f_{\mathcal{G}(j),x_{i(t')}}(w^{t'-K+j+1}),u_{w_{\mathcal{G}(j)}}^{t'}\right\rangle^2\right)\cdot\left\|u_{\mathcal{G}(k)}^{t'}\right\|^2\right]$$

$$\leq\sum_{k=0}^{K-1}\left[(d_k+4)\left\|\nabla f_{\mathcal{G}(k),x_{i(t')}}(w^{t'-K+k+1})\right\|^2+\sum_{j\neq k}\left(d_k\left\|\nabla f_{\mathcal{G}(j),x_{i(t')}}(w^{t'-K+j+1})\right\|^2\right)\right]$$

$$=\sum_{k=0}^{K-1}\left[d_k\sum_{j=0}^{K-1}\left\|\nabla f_{\mathcal{G}(j),x_{i(t')}}(w^{t'-K+j+1})\right\|^2+4\left\|\nabla f_{\mathcal{G}(k),x_{i(t')}}(w^{t'-K+k+1})\right\|^2\right]$$

$$=\left(\sum_{k=0}^{K-1}d_k\right)\sum_{j=0}^{K-1}\left\|\nabla f_{\mathcal{G}(j),x_{i(t')}}(w^{t'-K+j+1})\right\|^2+4\sum_{k=0}^{K-1}\left\|\nabla f_{\mathcal{G}(k),x_{i(t')}}(w^{t'-K+k+1})\right\|^2$$

$$=(d+4)\sum_{k=0}^{K-1}\left\|\nabla f_{\mathcal{G}(k),x_{i(t')}}(w^{t'-K+k+1})\right\|^2=(d+4)\left\|\sum_{k=0}^{K-1}\mathbf{I}_k\nabla f_{\mathcal{G}(k),x_{i(t')}}\left(w^{t'-K+k+1}\right)\right\|^2,$$

where the inequality is due to Lemma C.1 and C.2. $\square$

# D   proof of Lemma 5.6

*Proof.* For the convenience of analysis, we denote $t'=t-K+1$, then the update rule of algorithm 1 can be rewritten as

$$w_{\mathcal{G}(k)}^{t'+1}=w_{\mathcal{G}(k)}^{t'}-\gamma_{t'}\left(\hat{o}_L^{t'}\cdot u_{w_{\mathcal{G}(k)}}^{t'}\right)$$

Take expectation with respect to $u_{w_{\mathcal{G}(k)}}^{t'}$, we have

$$w_{\mathcal{G}(k)}^{t'+1}=w_{\mathcal{G}(k)}^{t'}-\gamma_{t'}\mathbb{E}_{u_{w_{\mathcal{G}(k)}}^{t'}}\left(\hat{o}_L^{t'}\cdot u_{w_{\mathcal{G}(k)}}^{t'}\right)=w_{\mathcal{G}(k)}^{t'}-\nabla f_{\mathcal{G}(k),x_{i(t')}}\left(w_{\mathcal{G}(k)}^{t'-K+k+1}\right)$$

Define diagonal matrices $\mathbf{I}_0, \cdots, \mathbf{I}_k, \cdots, \mathbf{I}_{K-1} \in \mathbb{R}^{d \times d}$ such that all the principle diagonal elements of $\mathbf{I}_k$ in $\mathcal{G}(k)$ are 1, and all the principle diagonal elements of $I_k$ in other than $\mathcal{G}(k)$ are 0. Then we have

$$\hat{o}_L^{t'} \cdot u_w^{t'} = \sum_{k=0}^{K-1} \mathbf{I}_k \cdot \hat{o}_L^{t'} \cdot u_{w_{\mathcal{G}(k)}}^{t'}$$

$$\nabla f_{x_{i(t')}}\left(w^{t'-K+k+1}\right) = \sum_{k=0}^{K-1} \mathbf{I}_k \nabla f_{\mathcal{G}(k), x_{i(t')}}\left(w_{\mathcal{G}(k)}^{t'-K+k+1}\right)$$

Since $f(\cdot)$ is $L$-smooth, the following inequality holds that:

$$f(w^{t'+1}) \leq f(w^{t'}) + \left\langle \nabla f(w^{t'}), w^{t'+1} - w^{t'} \right\rangle + \frac{L}{2}\|w^{t'+1} - w^{t'}\|^2$$

From the update rule of Algorithm 1, we take expectation with respect to all random variables on both sides and obtain:

$$\mathbb{E}[f(w^{t'+1})] \leq f(w^{t'}) - \gamma_{t'}\mathbb{E}\left[\nabla f(w^{t'})^{\mathsf{T}}\left(\sum_{k=0}^{K-1} \mathbf{I}_k \cdot \hat{o}_L^{t'} \cdot u_{w_{\mathcal{G}(k)}}^{t'}\right)\right] + \frac{L\gamma_{t'}^2}{2}\mathbb{E}\left\|\sum_{k=0}^{K-1} \mathbf{I}_k \cdot \hat{o}_{L_{K-1}}^{t'} u_{w_{\mathcal{G}(k)}}^{t'}\right\|^2$$

$$= f(w^{t'}) - \gamma_{t'}\sum_{k=0}^{K-1} \nabla f(w^{t'})^{\mathsf{T}}\mathbf{I}_k\left(\nabla f_{\mathcal{G}(k)}\left(w^{t'-K+k+1}\right) - \nabla f_{\mathcal{G}(k)}\left(w^{t'}\right) + \nabla f_{\mathcal{G}(k)}\left(w^{t'}\right)\right)$$

$$+ \frac{L\gamma_{t'}^2}{2}\mathbb{E}\left\|\sum_{k=0}^{K-1} \mathbf{I}_k \cdot \hat{o}_{L_{K-1}}^{t'} u_{w_{\mathcal{G}(k)}}^{t'} - \nabla f(w^{t'}) + \nabla f(w^{t'})\right\|^2$$

$$= f(w^{t'}) - \gamma_{t'}\left\|\nabla f(w^{t'})\right\|^2 - \gamma_{t'}\sum_{k=0}^{K-1} \nabla f(w^{t'})^{\mathsf{T}}\mathbf{I}_k\left(\nabla f_{\mathcal{G}(k)}\left(w^{t'-K+k+1}\right) - \nabla f_{\mathcal{G}(k)}\left(w^{t'}\right)\right)$$

$$+ \frac{L\gamma_{t'}^2}{2}\left\|\nabla f(w^{t'})\right\|^2 + \frac{L\gamma_{t'}^2}{2}\mathbb{E}\left\|\sum_{k=0}^{K-1} \mathbf{I}_k \cdot \hat{o}_{L_{K-1}}^{t'} \cdot u_{w_{\mathcal{G}(k)}}^{t'} - \nabla f(w^{t'})\right\|^2$$

$$+ L\gamma_{t'}^2\sum_{k=0}^{K-1} \nabla f(w^{t'})^{\mathsf{T}}\mathbf{I}_k\left(\nabla f_{\mathcal{G}(k)}\left(w^{t'-K+k+1}\right) - \nabla f_{\mathcal{G}(k)}\left(w^{t'}\right)\right)$$

$$= f(w^{t'}) - \left(\gamma_{t'} - \frac{L\gamma_{t'}^2}{2}\right)\left\|\nabla f(w^{t'})\right\|^2 + \underbrace{\frac{L\gamma_{t'}^2}{2}\mathbb{E}\left\|\sum_{k=0}^{K-1} \mathbf{I}_k \cdot \hat{o}_{L_{K-1}}^{t'} \cdot u_{w_{\mathcal{G}(k)}}^{t'} - \nabla f(w^{t'})\right\|^2}_{Q_1}$$

$$\underbrace{- (\gamma_{t'} - L\gamma_{t'}^2)\sum_{k=0}^{K-1} \nabla f(w^{t'})^{\mathsf{T}}\mathbf{I}_k\left(\nabla f_{\mathcal{G}(k)}\left(w^{t'-K+k+1}\right) - \nabla f_{\mathcal{G}(k)}\left(w^{t'}\right)\right)}_{Q_2},$$

Using the fact that $\|x + y\|^2 \leq 2\|x\|^2 + 2\|y\|^2$ and $xy \leq \frac{1}{2}\|x\|^2 + \frac{1}{2}\|y\|^2$, we have

$$Q_1 = \frac{L\gamma_{t'}^2}{2}\mathbb{E}\left\|\sum_{k=0}^{K-1} \mathbf{I}_k \cdot \hat{o}_{L_{K-1}}^{t'} \cdot u_{w_{\mathcal{G}(k)}}^{t'} - \nabla f(w^{t'}) - \sum_{k=0}^{K-1} \mathbf{I}_k \nabla f_{\mathcal{G}(k)}\left(w^{t'-K+k+1}\right)\right.$$

$$\left. + \sum_{k=0}^{K-1} \mathbf{I}_k \nabla f_{\mathcal{G}(k)}\left(w^{t'-K+k+1}\right)\right\|^2$$

$$\leq L\gamma_{t'}^2\underbrace{\mathbb{E}\left\|\sum_{k=0}^{K-1} \mathbf{I}_k \cdot \hat{o}_{L_{K-1}}^{t'} \cdot u_{w_{\mathcal{G}(k)}}^{t'} - \sum_{k=0}^{K-1} \mathbf{I}_k \nabla f_{\mathcal{G}(k)}\left(w^{t'-K+k+1}\right)\right\|^2}_{Q_3} +$$

$$+ L\gamma_{t'}^2 \underbrace{\left\| \sum_{k=0}^{K-1} \mathbf{I}_k \nabla f_{\mathcal{G}(k)}\left(w^{t'-K+k+1}\right) - \nabla f(w^{t'}) \right\|^2}_{Q_4}$$

$$Q_2 = -\left(\gamma_{t'} - L\gamma_{t'}^2\right) \sum_{k=0}^{K-1} \nabla f(w^{t'})^{\mathsf{T}} \mathbf{I}_k \left(\nabla f_{\mathcal{G}(k)}\left(w^{t'-K+k+1}\right) - \nabla f_{\mathcal{G}(k)}\left(w^{t'}\right)\right)$$

$$\leq \frac{\gamma_{t'} - L\gamma_{t'}^2}{2} \left\|\nabla f(w^{t'})\right\|^2 + \frac{\gamma_{t'} - L\gamma_{t'}^2}{2} \left\| \sum_{k=0}^{K-1} \mathbf{I}_k \nabla_{\mathcal{G}(k)} f(w^{t'-K+k+1}) - \nabla f(w^{t'}) \right\|^2$$

Using $\mathbb{E}\|\xi - \mathbb{E}[\xi]\|^2 \leq \mathbb{E}\|\xi\|^2$, we have

$$Q_3 = \mathbb{E} \left\| \sum_{k=0}^{K-1} \mathbf{I}_k \cdot \hat{o}_{L_{K-1}}^{t'} \cdot u_{w_{\mathcal{G}(k)}}^{t'} - \sum_{k=0}^{K-1} \mathbf{I}_k \nabla f_{\mathcal{G}(k)}\left(w^{t'-K+k+1}\right) \right\|^2$$

$$\leq \mathbb{E} \left\| \sum_{k=0}^{K-1} \mathbf{I}_k \cdot \hat{o}_{L_{K-1}}^{t'} \cdot u_{w_{\mathcal{G}(k)}}^{t'} \right\|^2$$

$$\leq (d+4) \left\| \sum_{k=0}^{K-1} \mathbf{I}_k \nabla f_{\mathcal{G}(k),x_{i(t')}}\left(w^{t'-K+k+1}\right) \right\|^2$$

$$= (d+4) \sum_{k=0}^{K-1} \left\| \nabla f_{\mathcal{G}(k),x_{i(t')}}(w^{t'-K+k+1}) \right\|^2$$

$$\leq (d+4)KM,$$

where the second inequality is due to Lemma 5.3. Then we bound $Q_4$,

$$Q_4 = \left\| \sum_{k=0}^{K-1} \mathbf{I}_k \nabla f_{\mathcal{G}(k)}\left(w^{t'-K+k+1}\right) - \nabla f(w^{t'}) \right\|^2 = \sum_{k=0}^{K-1} \left\| \nabla f_{\mathcal{G}(k)}(w^{t'-K+k+1}) - \nabla f_{\mathcal{G}(k)}(w^{t'}) \right\|^2$$

$$\leq \sum_{k=0}^{K-1} \left\| \nabla f(w^{t'-K+k+1}) - \nabla f(w^{t'}) \right\|^2$$

$$\leq L^2 \sum_{k=0}^{K-1} \left\| w^{t'} - w^{t'-K+k+1} \right\|^2$$

$$= L^2 \sum_{k=0}^{K-1} \left\| \sum_{j=\max\{0,t'-K+k+1\}}^{t'-1} (w^{j+1} - w^j) \right\|^2 = L^2 \sum_{k=0}^{K-1} \left\| \sum_{j=\max\{0,t'-K+k+1\}}^{t'-1} \gamma_j (\hat{o}_{L_{K-1}}^j \cdot u_w^j) \right\|^2$$

$$\leq L^2 \gamma_{\max\{0,t'-K+1\}}^2 \sum_{k=0}^{K-1} K \sum_{j=\max\{0,t'-K+k+1\}}^{t'-1} (d+4) \left\| \sum_{k=0}^{K-1} \nabla f_{\mathcal{G}(k),x_{(j)}}(w^{t'-K+k+1}) \right\|^2$$

$$\leq (d+4)KL\gamma_{t'} \frac{\gamma_{\max\{0,t'-K+1\}}}{\gamma_{t'}} \sum_{k=0}^{K-1} \sum_{j=\max\{0,t'-K+k+1\}}^{t'-1} \left\| \sum_{k=0}^{K-1} \nabla f_{\mathcal{G}(k),x_{(j)}}(w^{t'-K+k+1}) \right\|^2$$

$$\leq (d+4)L\gamma_{t'}\sigma K^4 M,$$

where the second inequality is from Assumption 5.1, the third inequality is due to Lemma 5.3, the fourth inequality follows from $L\gamma_{t'} < 1$, the last inequality follows from the inequality of arithmetic and geometric means, Assumption 5.2 and $\sigma := \max_{t'} \frac{\gamma_{\max\{0,t'-K+1\}}}{\gamma_{t'}}$. Integrating the upper bound together, we have

$$\mathbb{E}\left[f(w^{t'+1}) - f(w')\right] \leq -\frac{\gamma_{t'}}{2} \left\|\nabla f(w^{t'})\right\|^2 + (d+4)L\gamma_{t'}^2 KM + \frac{\gamma_{t'} + L\gamma_{t'}^2}{2}(d+4)L\gamma_{t'}\sigma K^4 M$$

$$\leq -\frac{\gamma_{t'}}{2}\left\|\nabla f(w^{t'})\right\|^2 + 2(d+4)L\gamma_{t'}^2(KM + \sigma K^4 M)$$

$$= -\frac{\gamma_{t'}}{2}\left\|\nabla f(w^{t'})\right\|^2 + 2(d+4)L\gamma_{t'}^2 M_K,$$

where we let $M_K = KM + \sigma K^4 M$.

$\square$

# E    Proof of Theorem 5.6

*Proof.* Let $\gamma_t = \gamma$ be a constant, taking total expectation in Lemma 5.5, we have

$$\mathbb{E}\left[f(w^{t'+1})\right] - \mathbb{E}\left[f(w^{t'})\right] \leq -\frac{\gamma}{2}\mathbb{E}\|\nabla f(w^{t'})\|^2 + 2(d+4)L\gamma^2 M_K,$$

where $\sigma = 1$ and $M_k = KM + K^4 M$. summing the above inequality from $t' = 0$ to $T - 1$ we have

$$\mathbb{E}[f(w^T)] - f(w^0) \leq -\frac{\gamma}{2}\sum_{t'=0}^{T-1}\mathbb{E}\|\nabla f(w^{t'})\|^2 + 2T(d+4)\gamma^2 L M_K$$

Then we have

$$\frac{1}{T}\sum_{t'=0}^{T-1}\mathbb{E}\|\nabla f(w^{t'})\|^2 \leq \frac{2(f(w^0) - f(w^*))}{\gamma T} + 4(d+4)L\gamma M_K.$$

$\square$

# F    Proof of Theorem 5.7

*Proof.* Let $\{\gamma_{t'}\}$ be a diminishing sequence and $\gamma_{t'} = \frac{\gamma_0}{t'+1}$, such that $\sigma < K$ and $M_K = KM + K^5 M$. Taking expectation in Lemma 5.5 and summing it from $t' = 0$ to $T - 1$, we have

$$\mathbb{E}[f(w^T)] - f(w^0) \leq -\frac{1}{2}\sum_{t'=0}^{T-1}\gamma_t\mathbb{E}\|\nabla f(w^{t'})\|^2 + \sum_{t'=0}^{T-1}2(d+4)\gamma_{t'}^2 L M_K.$$

Letting $\Gamma_T = \sum_{t'=0}^{T-1}\gamma_{t'}$, then we have

$$\frac{1}{\Gamma_T}\sum_{t'=0}^{T-1}\gamma_{t'}\mathbb{E}\|\nabla f(w^{t'})\|^2 \leq \frac{2\left(f(w^0) - f(w^*)\right)}{\Gamma_T} + \frac{\sum_{t'=0}^{T-1}4(d+4)\gamma_{t'}^2 L M_K}{\Gamma_T}$$

$\square$

# G    Details of AsyncFGD

## G.1    Working with Adam

We provide example for AsyncFGD working with Adam in Algorithm 2. Minimal changes are made on Adam by substituting the gradient the estimator using Forward Gradient.

## G.2    Execution Details

Details are presented in Figure 5. By pipelining over time dimension, we can preserve buffer for input in only one timestamp and still achieve parallel computation.

**Algorithm 2** AsyncFGD-Adam

**Initialize:** Stepsize sequence $\{\gamma_t\}_{t=K-1}^{T-1}$, weight $w^0 = \left[w_{\mathcal{G}(0)}^0, \cdots, w_{\mathcal{G}(K-1)}^0\right] \in \mathbb{R}^d, m_{\mathcal{G}(k)} = 0, v_{\mathcal{G}(k)} = 0, \beta_1 = 0.9, \beta_2 = 0.999, \eta = 1e-8$

1: **for** $t = 0, 1, \cdots, T-1$ **do**
2:     **for** $k = 0, 1, \cdots, K-1$ **in parallel do**
3:         Read $\hat{h}_{L_{k-1}}^{t-k}, \hat{o}_{L_{k-1}}^{t-k}$ from storage if $k \neq 0$
4:         Compute $\hat{h}_{L_k}^{t-k}, \hat{o}_{L_k}^{t-k}$
5:         Send $\hat{h}_{L_k}^{t-k}, \hat{o}_{L_k}^{t-k}$ to next worker's storage if $k \neq K-1$
6:     **end for**
7:     Broadcast $\hat{o}_{L_{K-1}}^{t-K+1}$
8:     **for** $k = 0, 1, \cdots, K-1$ **in parallel do**
9:         Compute $\Delta w_{\mathcal{G}(k)}^{t-K+1} = \hat{o}_{L_{K-1}}^{t-K+1} u_{w_{\mathcal{G}(k)}}^{t-K+1}$
10:       Update $m_{\mathcal{G}(k)} = \beta_1 \Delta w_{\mathcal{G}(k)}^{t-K+1} + (1-\beta_1)m_{\mathcal{G}(k)}$
11:       Update $v_{\mathcal{G}(k)} = \beta_2 \Delta w_{\mathcal{G}(k)}^{t-K+1} + (1-\beta_2)v_{\mathcal{G}(k)}$
12:       Compute $\hat{m}_{\mathcal{G}(k)} = m_{\mathcal{G}(k)}/\beta_1^t$
13:       Compute $\hat{v}_{\mathcal{G}(k)} = v_{\mathcal{G}(k)}/\beta_2^t$
14:       Update $w_{\mathcal{G}(k)}^{t-K+2} = w_{\mathcal{G}(k)}^{t-K+1} - \gamma_{t-K+1}\hat{m}_{\mathcal{G}(k)}/\hat{v}_{\mathcal{G}(k)}$
15:     **end for**
16: **end for**

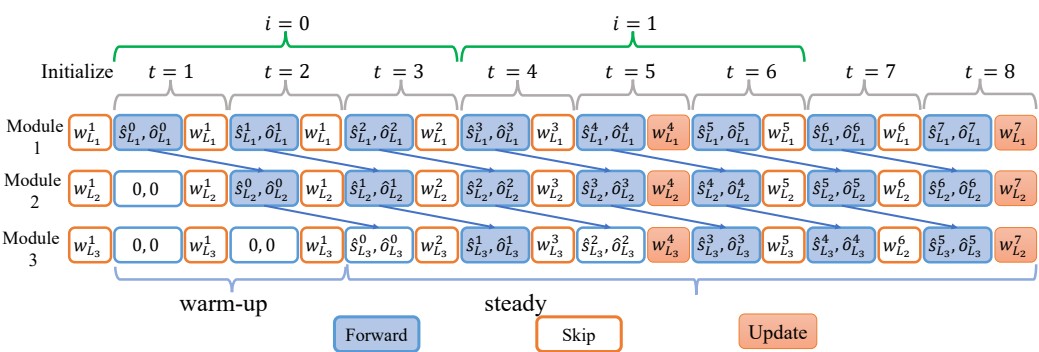

Figure 5: Details in executon of AsyncFGD-RNN with 3 modules. In the skip stage, only the host accumulate loss and its jvp value and other workers will jump right into the next state.

### G.3 Extension: AsyncFGD-Recursive

In this section, we extend the potential of AsyncFGD by exploring the parallelization of sequential inputs in RNNs with reduced memory footprint, necessitating the preservation of input for only a single timestamp.

We adopt a one-to-many RNN network for ease of illustration and denote the equal length of each sequence as $n$. We begin by refactoring the original loss for RNNs in terms of cumulative loss and new activation. Here, $s_l^t$ signifies the hidden state at timestamp $t$ on layer $l$. At timestamp $t$, each layer ingests $(s_{l-1}^t, s_l^{t-1})$ as input, generating $s_l^t = F_l(s_{l-1}^t, s_l^{t-1}, w_l)$. We represent the stacked latent states passed from $t-1$ as $s^{t-1} = [s_1^{t-1}, s_2^{t-1}, \ldots, s_L^{t-1}]$ and the output as $y_t = F(s_0^t, s^{t-1}; w)$, where $s_0^t$ symbolizes the input data $x_t$. The cumulative loss from timestamp $1 \sim T$ is given by:

$$\sum_{t=1}^{T} f(F(x_t, s^{t-1}; w), y_t) \tag{15}$$

We next refactor equation 2 for the $i_{th}$ sequential input in iteration $i, i \geq 0$ as:

$$w_l^{i+1} = w_l^i - \gamma_i \frac{\partial \mathcal{L}_i}{\partial w_l^i}, \quad \forall l \in 1, 2, \ldots, L \tag{16}$$

where $\mathcal{L}i := \sum t = in + 1^{(i+1)n} f(F(x_t, s^{t-1}; w), y_t)$ represents the loss for the $i_{th}$ sequence. We break the dependency between timestamps and iterations by employing dynamic staleness in AsynFGD. Specifically, the computation in module $k \in 1, 2, \cdots, K$ at timestamp $t$ is defined as follows:

$$\hat{s}L_k^{t-k} = f_k\left(\hat{s}L_{k-1}^{t-k}, \hat{s}\mathcal{G}(k)^{t-k-1}; w\mathcal{G}(k)^{t-2K+k+2}\right) \tag{17}$$

$$\hat{o}L_k^{t-k} = Jf_k\left(\hat{s}Lk-1^{t-k}, \hat{s}\mathcal{G}(k)^{t-k-1}; w\mathcal{G}(k)^{t-2K+k+2}\right) u_{\mathcal{G}(k)}^{t-k}, \tag{18}$$

$$\text{where } u_{\mathcal{G}(k)}^{t-k} = [\hat{o}Lk-1^{t-k\mathsf{T}}, \hat{o}\mathcal{G}(k)^{t-k-1\mathsf{T}}, uw_{\mathcal{G}(k)}^{t-k\mathsf{T}}]^{\mathsf{T}}$$

Given that tasks belonging to the same iteration use identical parameters, we use $\delta(k, t, i) = t - ni - k - 1, t \in [in+1, (i+1)n]$ to quantify this difference for the $i_{th}$ sequential. If $\delta(k, t, i) \leq 0$, then module $k$ uses stale parameters from iteration $i - 1$ at timestamp $t$. AsyncFGD-RNN only updates the parameter upon the completion of the last computation in the sequence. Specifically, we use:

$$w^{t-K+2} = \begin{cases} w^{t-K+1}, & \text{if } \frac{t-K+1}{n} \notin \mathbb{N}^* \\ w^{t-K+1} - \gamma_{\lfloor \frac{t-K}{n} \rfloor} \mathbb{E}uw^t\left(\left(\frac{\partial \mathcal{L}\lfloor \frac{t-K}{n} \rfloor}{\partial sL_K^{t-K}} o_{L_K}^{t-K}\right) u_w^{(t-K)}\right), & \text{otherwise} \end{cases}$$

Refer to figure 5 for detailed execution instructions. By combining training and prediction, we can process data from different timestamps of sequential input, maintain a buffer for just a single timestamp, and still achieve parallelization among various workers.

# H  Training Details

In this section, we explain some details in the trainning process.

## H.1  Random Seed

The seeds for all experiments are fixed to 0.

## H.2  Description of Network Architecture

### H.2.1  Models Deployed in Section 6.2.

The network structures of $ConvS, ConvL, FCS$ and $FCL$ are enumerated in Tables 8, 11, 9, and 10 correspondingly. In these designations, the subscripts $ReLU$ and $Tanh$ signify the particular activation function used in the model. Moreover, the larger models, denoted as counterparts, incorporate batch normalization layers for enhanced performance.

### H.2.2  Models used in Section 6.3.

We delve into the specific models employed in Section 6.3. For MobileNet, we utilize the structure of MobileNet_V3_Small. The ResNet-18 structure is used when implementing ResNet. The model EfficientNet_B0 signifies the EfficientNet architecture. The MNASNet0_5 is used for MNasNet. Lastly, we adopt ShuffleNet_V2_X0_5 for ShuffleNet.

## H.3 Model Splitting

In this section, we provide details for how to split the model into consective moduels and distribute them in different worker, we will first provide how to split models in Section 6.2, then we provide how to split model in 6.3.

### H.3.1 Model Splitting in Section 6.2

In Section 6.2, all models are split with $K = 3$. Details for how to split the ConvS, ConvL, FCS, FCL are repented in Table 5 Table 6, Table3 and Table 4, respectively.

Table 3: Details for model splitting for ConvS, definition for layers can be found in Table 8

| K | Layers |
|---|--------|
| 1 | conv1, act |
| 2 | pool1, fc1 |
| 3 | act2, fc2 |

Table 4: Details for model splitting for ConvL, definition for layers can be found in Table 11

| K | Layers |
|---|--------|
| 1 | conv1, bn1, act1, pool1, conv2, bn2, act2, pool2 |
| 2 | conv3, bn3, act3, pool3,conv4, bn4, act4, pool4 |
| 3 | conv5, bn5, act5, pool5, fc1 |

Table 5: Details for model splitting for FCS, definition for layers can be found in Table 9

| K | Layers |
|---|--------|
| 1 | fc1,ac1 |
| 2 | fc2,ac2 |
| 3 | fc3,ac3 |

Table 6: Details for model splitting for FSL, definition for layers can be found in Table10

| K | Layers |
|---|--------|
| 1 | fc1, bn1, ac1, fc2, bn2, ac2 |
| 2 | fc3, bn3, ac3 ,fc4, bn4, ac4 |
| 3 | fc5,bn5,ac5,fc5 |

### H.3.2 Model Splitting in Section 6.3

In section 6.3, all models are divided into four parts ($K = 4$). Detailed descriptions of how each model is split are provided below. Note that 'head' and 'tail' refer to the layers before and after the main blocks of each architecture, respectively, which are assigned to the first and the last worker:

- **ResNet-18:** The core of ResNet-18 consists of 4 Residual Blocks, each distributed to one of the four workers.

- **EfficientNet:** The core of EfficientNet consists of 7 Mobile Inverted Bottlenecks (MBConv). The first worker handles MBConv 1, the second handles MBConv 2 to 3, the third manages MBConv 4 to 6, and the last one manages MBConv 7.

- **MoblieNet:** The core of MoblieNetV3-small includes 13 layers of bottlenecks. The first worker handles layers 1 to 3, the second manages layers 4 to 7, the third manages layers 8 to 11, and the last worker handles layers 12 to 13.

- **MnasNet:** The core of MnasNet consists of 6 blocks of inverted residuals. Blocks 1 to 2, 3 to 5, and 6 are assigned to workers 2, 3, and 4 respectively, while the first worker only handles the head.

- **ShuffleNet:** The core of ShuffleNet consists of 3 stages, each assigned to workers 2, 3, and 4, respectively.

# I  Additional Experimental Results

## I.1  Ablation study in $\alpha$

We have incremented the value of $\alpha_{bias}$ gradually, with a step size of 0.0075, over 20 epochs. This process can be generalized using the following equations:

$$
\alpha_{bias} = \begin{cases} t \times \dfrac{\alpha^*_{bias}}{20}, & t <= 20 \\ \alpha^*_{bias}, & \text{otherwise} \end{cases}
$$

Here, $\alpha^*_{bias}$ control the rate of increase and the maximum attainable value of $\alpha_{bias}$, respectively. The ablation study with respect to $\alpha^*_{bias}$ is presented in Table 7.

We observe that reducing $\alpha^*_{bias}$ to 0, which corresponds to only updating the classifier, still results in performance gains compared to updating the full model. This improvement can be attributed to reduced variance. As $\alpha^*_{bias}$ increases, we generally see better results, since the norm of the gradient approximation increases. However, when $\alpha^*_{bias}$ exceeds 0.25, we sometimes observe a performance drop due to the corresponding increase in variance.

Table 7: Ablation study on different value of $\alpha^*_{bias}$

| Dataset | Model | $\alpha^*_{bias}$ | | | | | | | |
|---|---|---|---|---|---|---|---|---|---|
| | | 0.00 | 0.03 | 0.06 | 0.09 | 0.12 | 0.15 | 0.20 | 0.25 |
| CIFAR10 | Res-18 | 0.838 | 0.838 | 0.845 | 0.855 | 0.865 | 0.878 | 0.872 | 0.885 |
| | Mobile | 0.898 | 0.912 | 0.911 | 0.910 | 0.913 | 0.911 | 0.914 | 0.909 |
| | Efficient | 0.892 | 0.900 | 0.902 | 0.903 | 0.902 | 0.902 | 0.887 | 0.895 |
| | Shuffle | 0.788 | 0.805 | 0.808 | 0.812 | 0.820 | 0.820 | 0.822 | 0.825 |
| | Mnas | 0.782 | 0.790 | 0.788 | 0.789 | 0.788 | 0.789 | 0.777 | 0.782 |
| FMNIST | Res-18 | 0.866 | 0.869 | 0.871 | 0.873 | 0.875 | 0.880 | 0.882 | 0.884 |
| | Mobile | 0.890 | 0.908 | 0.906 | 0.906 | 0.906 | 0.906 | 0.899 | 0.901 |
| | Efficient | 0.889 | 0.904 | 0.906 | 0.902 | 0.905 | 0.904 | 0.908 | 0.897 |
| | Shuffle | 0.849 | 0.854 | 0.857 | 0.860 | 0.864 | 0.870 | 0.870 | 0.877 |
| | Mnas | 0.854 | 0.868 | 0.870 | 0.870 | 0.870 | 0.870 | 0.864 | 0.864 |

## I.2  Acceleration across Various Platforms and Architectures

In Section 6.5, we examined the acceleration of AsyncFGD in comparison to vanilla FGD on ResNet-18, using two hardware platforms: 1) NVIDIA AGX Orin, an embedded device, and 2) a cluster of four NVIDIA 1080 Ti GPUs. These platforms were chosen to reflect real-world edge device scenarios and to simulate situations of ample computational power, such as in the case of stacked chips.

In this section, we expand our scope of investigation by incorporating two additional devices: 1) NVIDIA A100, and 2) Intel(R) Xeon(R) CPU E5-2678 v3 @2.50GHZ. These additions allow us to further examine acceleration under various conditions. We also provide supplementary results on acceleration with respect to different batch sizes to reflect variable input streams. Moreover, to emulate streamlined input, the mini-batch size of the synchronized pipeline is set to 1.

The performance of ResNet-18 with different batch sizes on the four NVIDIA 1080Ti GPUs, A100, and AGX Orin platforms is illustrated in Figures 6, 7, and 8, respectively. Results for MobileNetV3-small on AGX Orin are presented in Figure 10. A notable observation is that AsyncFGD performance appears largely insensitive to batch size. In contrast, other algorithms typically exhibit poorer performance with smaller batch sizes. Particularly, when the batch size is reduced to 1, these algorithms offer negligible performance improvements over vanilla FGD. Furthermore, the overall

acceleration on a single device is constrained by computational power. For instance, while AsyncFGD achieves a speedup of 2.84× on a four GPU cluster, it only delivers a 2.11 × speedup on a single AGX Orin. Communication also imposes a limit on the overall acceleration, as demonstrated by the superior performance on the A100 in comparison to the four-GPU cluster. This is attributable to the elimination of communication overhead on a single device, except for the sending and receiving operations of CUDA kernels.

Results for MobileNetV3-small with different batch sizes on CPU are depicted in Figure 9. Due to the inherently sequential execution pattern of CPUs, the acceleration is constrained, resulting in only modest speedup and advantage over other algorithms.

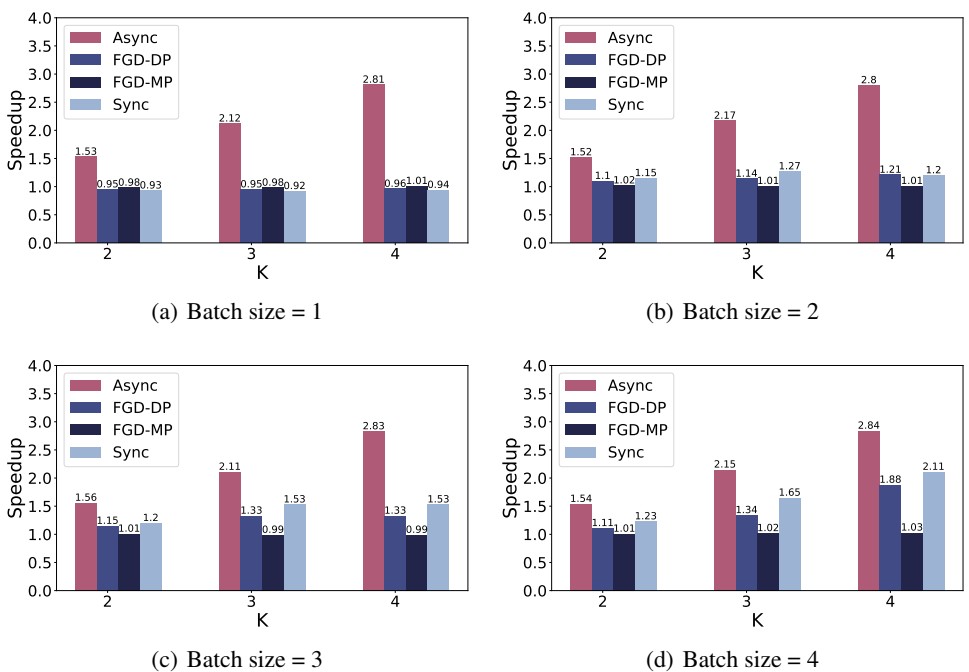

Figure 6: Acceleration with differernt batch size on ResNet-18, cluster with 4 Nvidia 1080 Ti

## I.3 Memory Profiling on Other Basic Units of Convolutional Neural Networks

This section outlines memory profiling for basic units within a Convolutional Neural Network (CNN). Commonly, a CNN layer is coupled with a batch normalization layer and an activation layer using ReLU, so we've combined these elements for our memory testing. We examine the memory consumption against the number of layers and present the results in Figure 11(a). For further examination, we also assess the memory consumption against the number of output channels and batch size, with results shown in Figures 11(b) and 11(c), respectively.

Our findings reveal that implementing forward gradients can significantly reduce memory consumption. Generally, the majority of memory usage in CNNs stems from intermediate results, since CNNs often operate in a 'broadcast then product' pattern (to be specific, they are referred as 'img2col'). Consequently, the additional memory required by the random tangent in AsyncFGD is minimal. As such, the memory consumption appears to be invariant to the number of layers, mainly because in the forward pass we discard almost all the intermediate variables.

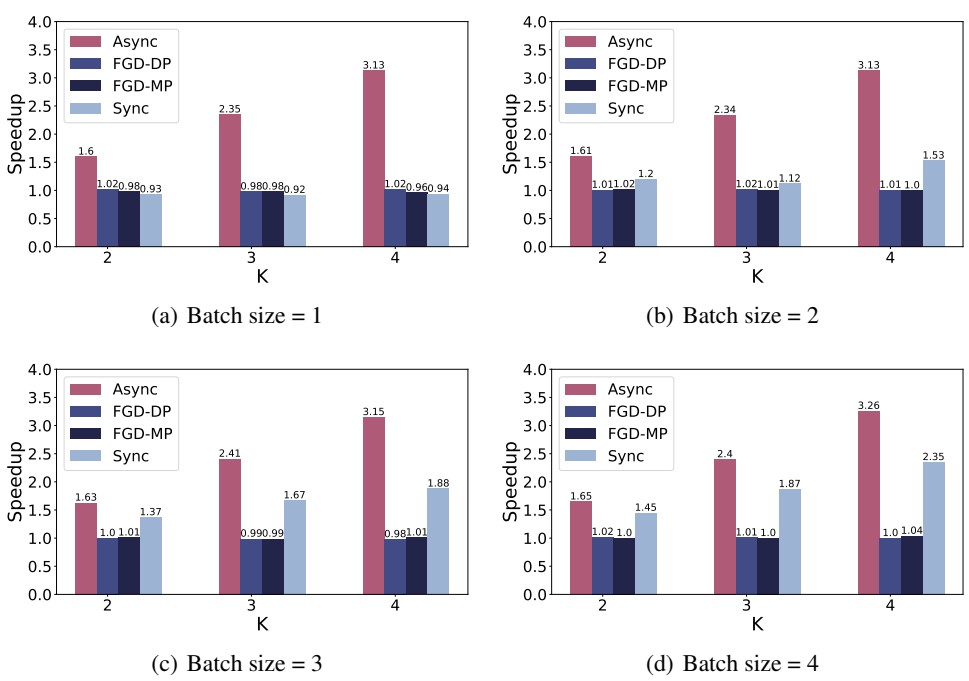

Figure 7: Acceleration with differernt batch size on ResNet-18, A100

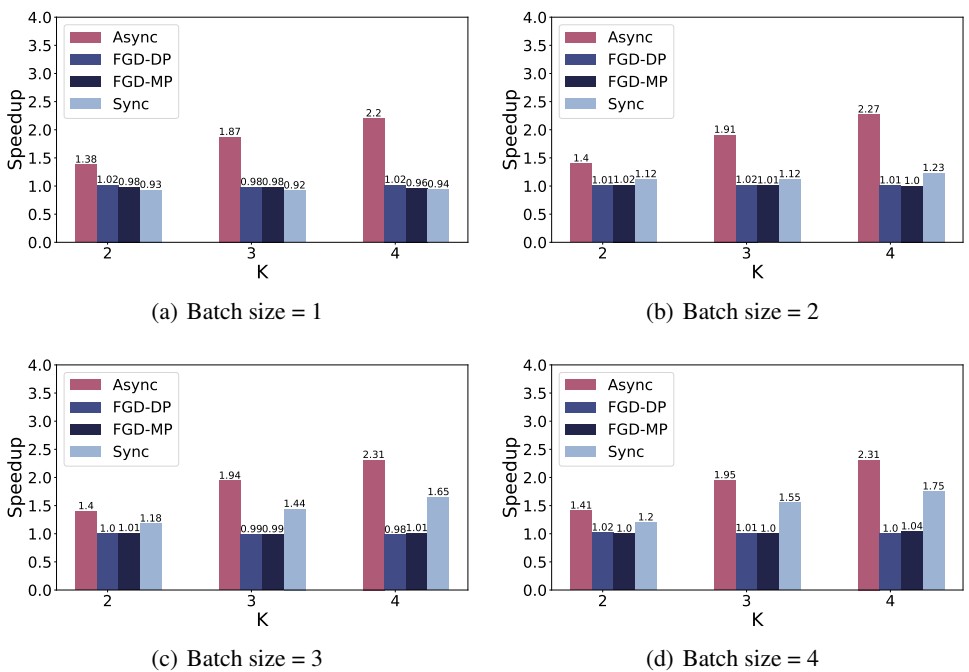

Figure 8: Acceleration with differernt batch size on ResNet-18, AGX Orin

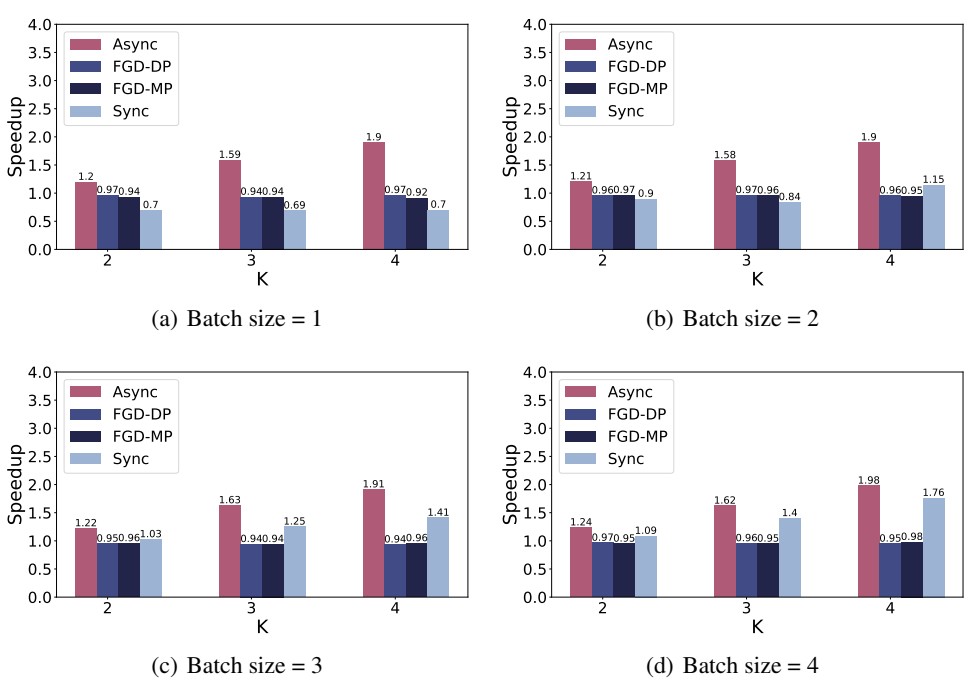

Figure 9: Acceleration with differernt batch size on MobileNetV3-small, Intel(R) Xeon(R) CPU E5-2678 v3 @ 2.50GHz

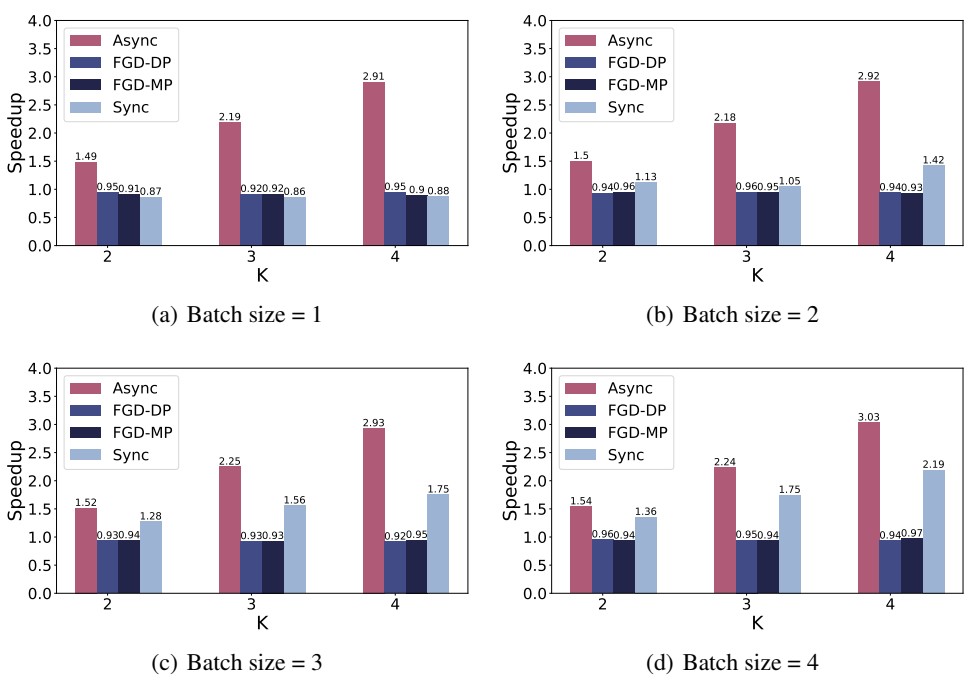

Figure 10: Acceleration with differernt batch size on MobileNetV3-small, AGX Orin

Table 8: Network architecture for $ConvS_{ReLU}$ and $ConvS_{Tanh}$. $ConvS_{ReLU}$ denotes using ReLU for the activation functions and $ConvS_{Tanh}$ denotes using Tanh as activation functions

| Layer | Type | Params |
|-------|------|--------|
| conv1 | Conv2d | out_channels=32, kernel_size=5, stride=1, padding=2 |
| act1 | ReLU/Tanh | N/A |
| pool1 | Maxpooll2d | kernel_size=2, stride=2, padding=0, dilation=1 |
| fc1 | Linear | out_features=1000 |
| act2 | ReLU/Tanh | N/A |
| fc2 | Linear | out_features=10 |

Table 10: Network architecture for $FCL_{ReLU}$ and $FCL_{Tanh}$. $FCL_{ReLU}$ denotes using ReLU for the activation functions and $FCL_{Tanh}$ denotes using Tanh as activation functions

| Layer | Type | Params |
|-------|------|--------|
| flatten | Flatten | N/A |
| fc1 | Linear | out_features=1024 |
| bn1 | BatchNorm1d | N/A |
| ac1 | ReLU/Tanh | N/A |
| fc2 | Linear | out_features=1024 |
| bn2 | BatchNorm1d | N/A |
| ac2 | ReLU/Tanh | N/A |
| fc3 | Linear | out_features=1024 |
| bn3 | BatchNorm1d | N/A |
| ac3 | ReLU/Tanh | N/A |
| fc4 | Linear | out_features=1024 |
| bn4 | BatchNorm1d | N/A |
| ac4 | ReLU/Tanh | N/A |
| fc5 | Linear | out_features=512 |
| bn5 | BatchNorm1d | N/A |
| ac5 | ReLU/Tanh | N/A |
| fc6 | Linear | out_features=10 |

Table 9: Network architecture for $FCS_{ReLU}$ and $FCS_{Tanh}$. $FCS_{ReLU}$ denotes using ReLU for the activation functions and $FCS_{Tanh}$ denotes using Tanh as activation functions

| Layer | Type | Params |
|-------|------|--------|
| flatten | Flatten | N/A |
| fc1 | Linear | out_features=1024 |
| ac1 | ReLU/Tanh | N/A |
| fc2 | Linear | out_features=512 |
| ac2 | ReLU/Tanh | N/A |
| fc3 | Linear | out_features=256 |

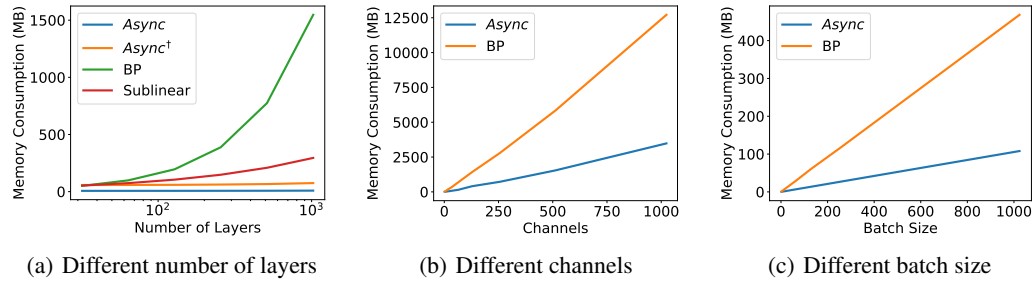

| (a) Different number of layers | (b) Different channels | (c) Different batch size |

Figure 11: Memory consumption of basic units in convoluntional networks, batch size=64, channels=3 and number of layers=18 unless appears as in the x axis

Table 11: Network architecture for $ConvL_{ReLU}$ and $ConvL_{Tanh}$. $ConvL_{ReLU}$ denotes using ReLU for the activation functions and $ConvL_{Tanh}$ denotes using Tanh as activation functions

| Layer | Type | Params |
|---|---|---|
| conv1 | Conv2d | out_channels=32, kernel_size=3, stride=1, padding=2 |
| bn1 | BatchNorm2d | N/A |
| act1 | ReLU/Tanh | N/A |
| pool1 | Maxpooll2d | kernel_size=2, stride=2, padding=0, dilation=1 |
| conv2 | Conv2d | out_channels=64, kernel_size=3, stride=1, padding=2 |
| bn2 | BatchNorm2d | N/A |
| act2 | ReLU/Tanh | N/A |
| pool2 | Maxpooll2d | kernel_size=2, stride=2, padding=0, dilation=1 |
| conv3 | Conv2d | out_channels=128, kernel_size=3, stride=1, padding=2 |
| bn3 | BatchNorm2d | N/A |
| act3 | ReLU/Tanh | N/A |
| pool3 | Maxpooll2d | kernel_size=2, stride=2, padding=0, dilation=1 |
| conv4 | Conv2d | out_channels=256, kernel_size=3, stride=1, padding=2 |
| bn4 | BatchNorm2d | N/A |
| act4 | ReLU/Tanh | N/A |
| pool4 | Maxpooll2d | kernel_size=2, stride=2, padding=0, dilation=1 |
| conv5 | Conv2d | out_channels=512, kernel_size=3, stride=1, padding=2 |
| bn5 | BatchNorm2d | N/A |
| act5 | ReLU/Tanh | N/A |
| pool5 | Maxpooll2d | kernel_size=2, stride=2, padding=0, dilation=1 |
| flatten | Flatten | N/A |
| fc1 | Linear | out_features=10 |

