# OpenReview forum: "Accelerated On-Device Forward Neural Network Training with Module-Wise Descending Asynchronism"
_NeurIPS.cc/2023/Conference — NeurIPS 2023 poster_

### Official Review · Reviewer_bhPv · 2023-06-26

**Soundness:** 2 fair
**Presentation:** 1 poor
**Contribution:** 2 fair
**Rating:** 5
**Confidence:** 4

**Summary:**

The paper focuses on enabling training on memory-constrained platforms. Instead of using conventional backpropagation-based methods, the proposed approach is based on forward gradient descent (FGD), which approximates the gradients through only forward passes. The paper claims that a brute-force utilisation of FGD leads to resource underutilisation in memory-constrained devices. To counteract this, the work proposes AsyncFGD, a method that enables multiple parallel workers to process different parts of the network using different samples, while allowing varying levels of staleness in the parameters of these parts.

**Strengths:**

1) Reducing the memory requirements of the training stage is a key and still open challenge towards enabling on-device training. In this context, FGD-based algorithms constitute an interesting approach that is still largely unexplored.
2) The paper provides both theoretical analysis and empirical evaluation of the proposed approach.

**Weaknesses:**

1) In the motivation of work, the paper makes a number of arguments that are hard to justify. In Section 2.2, the paper claims that "in edge devices with limited computational power, potential idle workers due to synchronization in synchronous pipelines like [12] are not optimal." However, this is rarely, or not at all, the case. Specifically, edge devices typically utilise their processing resources fully by exploiting the intra-layer parallelism, *e.g.* by performing convolutions or matrix multiplications. For the processor of such platforms to remain underutilised, either the device lies more towards server-grade or the target model is very lightweight, with small layers that contain minimal parallelism. This could make more sense in server settings, where layer pipelining has already been proposed (GPipe, PipeDream) as also mentioned in the paper. Therefore, the argument about Forward Locking and the serialisation of computations in the forward pass of FGD-based methods does not seem to be relevant for memory-constrained edge devices, where layer pipelining is hardly used during training.
As such, there are important questions with respect to the suitability of AsyncFGD. The presented implementation uses multiple workers to process different parts of the network, *i.e.* it employs module pipeling. For this to make sense as a strategy, the computational resources of the target device must be underutilised in the first place, *i.e.* when module pipelining is not used. In this context, what about batching? Batch processing is not touched upon in the paper. Why not have different workers processing different samples from the same batch, thus not requiring any stale parameters?
2) In Section 4.3, the paper makes a claim with respect to the attainable speedup. Specifically, it states that "It is easy to see that AsyncFGD can fully utilize the computation resources, thus achieving considerable speedup". This statement is very strong and cannot be supported solely by Fig. 2. In Fig. 2, we see a simple performance model for AsyncFGD, which indicates the potential acceleration *under certain conditions*. These conditions include (among others) *i)* whether the processor is really underutilised, *i.e.* whether there are really free cycles to have more than one module running, and *ii)* whether the multiple workers become external memory bandwidth-bound, in which case AsyncFGD can lead to slowdown - this is very common in edge devices where multiple components of the target chip share the same bandwidth to the external memory. In the scope of this work that aims at memory-constrained devices, the assumption that the memory bandwidth is abundant contradicts the whole setup. Overall, the logic of the paper with respect to acceleration leaves out many important considerations.
3) In Section 6.6, one would expect the speedup to plateau or even lead to slowdown, when *K* increases above a certain point, where the workers would start to context-switch excessively. How is *K* selected? What exactly affects it? Both the model architecture, the target dataset and the target platform would play important roles (see also point 2 above on this). In its current version, the paper is missing a thorough investigation of this fact.
4) In Section 6.2, while the objective of the experiments and the selection of baselines are sound, the network architectures are arbitrary and simple. It is hard to draw conclusions on the efficacy of AsyncFGD.
5) In Section 6.4, it is not clear what the setup is. The models are pre-trained on ImageNet and then fine-tuned on a given dataset using one of the algorithms in Table 2. Why does AsyncFGD* give higher accuracy than FGD and plain AsyncFGD? Furthermore, even Async* falls short compared to the BP-based baseline. Especially, given the issues of point 1) above, it is not evident that AsyncFGD has useful merits or under which conditions it would be a better choice than competing methods.
6) In Section 6.5, the memory footprint comparison between FGD and AsyncFGD is missing in Fig. 3a and b - how do they compare? Importantly, how does the selection of *K* affect the memory footprint? Having more parallel workers can have a significant impact on memory consumption, through thread bookkeeping and cache misses.
7) As specified in the Appendix, only a single random seed is used in the experiments.

**Questions:**

1) Is something accidentally missing from Section 6.1?
2) Please also see points 5) and 6) above for things that were not clear.
3) Please proof-read the paper as there are some typos.


**Limitations:**

The paper does not explicitly discuss its limitations.

---

> ### Author Rebuttal · Authors · 2023-08-10
>
> Dear Reviewer bhPv,
>
> Thank you for your comprehensive review and insightful comments. We have carefully considered your feedback and offer the following responses:
>
> **Weaknesses:**
>
> **Q1:** "Does idle worker really exists when training on Edge device? If so, why not batching inputs to parallel computation among workers ?"
>
> **A1:** Your insightful question touches on two critical dimensions of efficient training on edge devices: the existence of idle workers and the potential for parallel computation through batching. Let's delve into these aspects:
>
> 1. **Existence of Idle Workers:** Our method, although not a one-size-fits-all solution, can specifically addresses scenarios where edge devices, despite unused computational power, face memory constraints with BP-based algorithms, i.e. mismatch for parallisim and memory consumption. Specifically, temporal models like RNNs optimized by BPTT or SNNs optimized by Spatio-temporal backpropagation (STBP), can be highly memory-intensive. On edge devices, this often leads to reaching memory limits before fully utilizing all computational resources, resulting in idle workers, for example, on AGX Orin, which is employed in our experiment,  its typical when RNN or SNN dealing inputs with more than 300 timestamps, the overall utilization of computing power only takes 40% without any spare memory to further utilize the idle workers. As emphasized in [1], the substantial memory demands pose a challenge in implementing RNNs on edge devices. AsyncFGD, our solution, employs Forward Gradient to relax memory constraints, thus fully utilizing otherwise idle workers.
> 2. **Batch Processing Considerations**: We concur with your insights on the merits of batch processing. While it is effective for historical data, there exist real-time scenarios, especially when edge devices are interfaced with specific sensors, that necessitate sequential data processing:
>
>    - **Real-time Adaptation and Streaming Data Learning**: Edge devices might need to swiftly adapt to and learn from newly acquired data in real-time fashion [2]. Moreover, when processing sequential information with temporal connections, as collected directly by sensors [3, 4], batching could interfere with these inherent relationships, disrupting the learning process [5].
>    - **Data Privacy and Security**: For applications dealing with sensitive data, storing it for batching could introduce privacy and security vulnerabilities. Immediate processing, followed by data disposal, can mitigate these risks.
>    - **Energy Efficiency**: Given the energy constraints typical of edge devices, batching can sometimes be more energy-intensive due to the demands of memory storage and data retrieval. In contrast, processing data as it arrives can be more energy-efficient in specific scenarios.
>
> In summary, we view our method as a viable solution for specific situations, particularly when edge devices face memory constraints with BP-based algorithms. However, we acknowledge that it may not be suitable for all scenarios. Your valuable feedback is instrumental in refining our approach for the revised manuscript.
>
> **Q2:** "Claims about Attainable Speedup ignore certain considerations."
>
> **A2:** We recognize your feedback on our speedup claims, especially in the context of bandwidth and computing power bottlenecks. In our experiments, we demonstrated positive results on AGX Orin, an edge device used in applications such as autonomous driving, robotics, and drones [7]. AGX Orin's off-chip bandwidth supports enough throughput, and its on-chip bandwidth facilitates communication between different workers on the chip. These features align with our AsyncFGD method, allowing for maximum utilization of computing resources and achieving acceleration. Given the broad definition and varied capabilities of edge devices, we acknowledge that our method might not achieve speedup on IoT-targeted chips, where bandwidth and computing power are restricted to even deploy a deep learning model. However, we believe our method has its merits in certain contexts, especially considering the rapid advancements in edge device's bandwidth, as evidenced by High Bandwidth Memory FPGAs. In light of your practical considerations, we will refine our claims and narrow the scope of our research in the revised manuscript. Your insights have been instrumental, and we appreciate your constructive feedback.
>
> **Q3:** "How is $K$ selected, considering model architecture, dataset, and platform?"
>
> **A3:** We appreciate your inquiry about the selection of $K$. Our extra experiments, detailed in Table 3 of the rebuttal PDF, confirm that as $K$ increases, the overheads can outweigh the benefits, leading to a plateau or even slowdown. This often happens when $K > 4$. Thus, the choice of $K$ requires careful consideration of the model's complexity, data characteristics, and platform constraints and we will add more experiments in the revised manuscript. Thank for your observation that has guided us to refine our work.
>
> **Q4:** Section 6.2's simple, arbitrary network architectures hinder AsyncFGD conclusions.
>
> **A4:** We apologize for any confusion. For comparisons with other BP-free algorithms, we intentionally used simpler network architectures in Section 6.3, as architectures used in the original papers are rather simple (for example, 2$\times$ 1000 FC layers in DFA [8] and two convolutional layers with 64 and 256 channels with $3 \times 3$ kerne in DRTP [9]). For efficient transfer learning in Table 2, and Fig 3(c), we selected widely-accepted lightweight models to represent a broader range of efficient models suitable for edge devices.
>
> We have reached the page limit. Please refer to the global author rebuttal for the remainng.

---

> > ### Comment · Reviewer_bhPv · 2023-08-14
> >
> > Thanks to the authors for their detailed response. Please make sure to integrate your answers to the main paper, so that its scope is more clearly defined.
> >
> > A few further comments:
> > **A1.1:** While it is true that memory-bound workloads (*e.g.* RNNs) can leave the computational resources underutilised due to the lack of enough main memory bandwidth, for the computational resources to perform another piece of work, unless there is data reuse in place, data still have to be transferred from the main memory to the processor.
> > **A1.2:** Batch processing can also be applied in real-time applications, depending on the sensor's data generation rate and the processing rate of the system. For instance, for a camera with a capture rate of 60 FPS and an application that requires 30 FPS for good user experience, the processing system could potentially use a batch size of 2.

---

> > > ### Author Response · Authors · 2023-08-16
> > >
> > > Dear Reviewer bhPv,
> > >
> > > Thank you for recognizing our efforts and for your insightful comments. We truly appreciate your engagement with our work, and we would like to further clarify some points to address your concerns and any misunderstandings from our previous rebuttal.
> > >
> > > **For A1.1:** We apologize for any misunderstanding in our previous communication. When referring to 'limited memory' in A1.1, we were highlighting the constraints on memory capacity, not the bandwidth. Indeed, there can be a mismatch between bandwidth and capacity, as evidenced by Intel® Stratix® 10 HBM2 DRAM with 256GB/sec bandwidth but only 4GB or 8GB density, or the 25.6GB/sec 4GB memory on Nvidia Jetson Nano. This is particularly relevant for edge devices with deep learning accelerators, such as Nvidia Jetson Orin NX, which offers 8GB capacity with a theoretical peak memory bandwidth of 102GB/sec. In such scenarios, an Out Of Memory (OOM) problem may arise when optimizing a sequential model, leaving workers and memory bandwidth underutilized.
> > >
> > > Our AsyncFGD, by utilizing forward gradient and relaxing constraints on capacity, enables full utilization of potential workers and bandwidth. While a perfect alignment of memory capacity, bandwidth, and computing power may limit AsyncFGD's application, mismatches often occur. Specifically, when memory consumption for optimizing a model breaks the balance, our method shines by discarding intermediate variables during the forward pass, achieving relatively low memory consumption and full utilization of the other two components.
> > >
> > > **For A1.2:** You are correct in noting the use of batch processing in real-time scenarios. As a complement to your observation, we'd like to highlight that many applications require the preservation of temporal connections in data collected by sensors, where batching could be less suitable.
> > >
> > > For example, **real-time anomaly detection** in fields like industrial monitoring, healthcare, and security systems often need to take special care for temporal connection and utilize sequential models on edge device. Examples include a multi-scale convolutional recurrent encoder-decoder in [1] or LSTM in [2].
> > >
> > > Similarly, **temporal pattern recognition**, where patterns emerge from both spatial and temporal information, also requires take care of temproal connection from data collected by sensors. Human activity recognition from spatio-temporal features [3] or driving behavior recognition using smartphone sensor data [4] are cases where temporal connections along the time dimension are vital.
> > >
> > > Additonally, **time-series forecasting**, in field like weather forecasting, stock market prediction, energy consumption prediction, utilizing temporal connections in sensor data for predictive modeling and forecasting future values [5, 6].
> > >
> > > In these scenarios, batching consecutive 'frames' and sending them to different workers may not be suitable, as it could disturb the temporal connection. Our method is designed to handle such sequential data naturally with a pipeline fasion, preserving its temporal connection while achieving acceleration, as detailed in the Appendix of our original paper.
> > >
> > > We sincerely thank you again for helping us identify more clearly the scenarios where our algorithm can be applied. Your feedback has been invaluable in refining our work, and we look forward to any further insights you may have.
> > >
> > > Warm regards
> > >
> > > [1] Zhang C, Song D, Chen Y, et al. A deep neural network for unsupervised anomaly detection and diagnosis in multivariate time series data[C]//Proceedings of the AAAI conference on artificial intelligence. 2019, 33(01): 1409-1416.
> > >
> > > [2] Sivapalan G, Nundy K K, Dev S, et al. ANNet: a lightweight neural network for ECG anomaly detection in IoT edge sensors[J]. IEEE Transactions on Biomedical Circuits and Systems, 2022, 16(1): 24-35.
> > >
> > > [3] Medina M Á L, Espinilla M, Paggeti C, et al. Activity recognition for iot devices using fuzzy spatio-temporal features as environmental sensor fusion[J]. Sensors (Basel, Switzerland), 2019, 19(16).
> > >
> > > [4] Zhang J, Wu Z, Li F, et al. Attention-based convolutional and recurrent neural networks for driving behavior recognition using smartphone sensor data[J]. IEEE Access, 2019, 7: 148031-148046.
> > >
> > > [5] Zhang Y F, Fitch P, Thorburn P J. Predicting the trend of dissolved oxygen based on the kPCA-RNN model[J]. Water, 2020, 12(2): 585.
> > >
> > > [6] Koppe G, Guloksuz S, Reininghaus U, et al. Recurrent neural networks in mobile sampling and intervention[J]. Schizophrenia bulletin, 2019, 45(2): 272-276.

---

### Official Review · Reviewer_yH2P · 2023-07-06

**Soundness:** 2 fair
**Presentation:** 3 good
**Contribution:** 2 fair
**Rating:** 5
**Confidence:** 5

**Summary:**

ThIS paper introduces AsyncFGD, a forward-only based training method. The key idea behidn this paper is integrating asynchronous updates with Forward Gradient Descent (FGD). The authors test their method on several small-scale datasets.

**Strengths:**

- The proposed proof of the convergence guarantee of AsyncFGD-SGD is new.
- According to the experiment results, this framework achieves a good acceleration rate on small-scale datasets.

**Weaknesses:**

- The novelty of this approach is not introduced clearly. There is no fundamental difference between the original forward-only training methods by introducing a pipeline mechanism among different layers and this methods.
- The experimental results are only shown on small-scale datasets.
- The comparison with other works is not sufficient [1].
- The proposed method has on-par memory consumption with BP based training yet with lower accuracy. The memory cost is crucial for some edge devices.

[1] Krithivasan S, Sen S, Venkataramani S, et al. Accelerating DNN Training Through Selective Localized Learning[J]. Frontiers in Neuroscience, 2022, 15: 759807.

**Questions:**

- Can the author explain the fundamental difference between the original forward-only training methods by introducing a pipeline mechanism among different layers and this methods?
- Can the author show the large-scale dataset experiments?

**Limitations:**

Please refer to the weakness sections.

---

> ### Author Rebuttal · Authors · 2023-08-10
>
> Dear Reviewer yH2P,
>
> Thank you for your comprehensive review and the valuable insights you provided. We genuinely appreciate your feedback, as it offers us a clear direction for refining our work. We have taken the time to address each of your concerns in detail.
>
> **Weaknesses:**
>
> **Q1:** "Unclear novelty and distinction from original forward-only training Methods with pjipeline parallelism."
>
> **A1:**  Thank you for your insightful question. Our method's novelty lies in its design to detach dependencies in each module during the forward pass, most importantly, with theoretical grantee and performance comparable to the original version. Specifically:
>
> - Dependencies among layers are preserved within each iteration.
> - The staleness of each module is bounded on a module-wise basis.
>
> Our approach's primary goal is to ease the dependency in the forward pass from an iterative optimization standpoint. Supported by our mathematical proof, it converges to critical points under non-convex conditions. Though the asynchronous pipeline parallelism that results from this relaxed constraint may seem similar to merely adding pipeline parallelism to forward gradient, it stems from a fundamentally different motivation and design philosophy. Moreover, it is worth noting that our method is the first to provide asynchronous acceleration on FGD.
>
> **Q2:** "Experimental results focus on small-scale datasets."
>
> **A2:** We recognize your point regarding the scale of our datasets. To address this, we have conducted additional experiments on ImageNet, ILSVRC2012 datasets. While training the model from scratch led to a gap in accuracy compared to BP, we found that in online training scenarios, efficient transfer learning strategy can be employed to reduced this difference. As detailed in Table 2 of the rebuttal PDF, our AsyncFGD method, when combined with this strategy (Async$^*$), outperformed other BP-free algorithms on these larger datasets, approaching closer to BP's accuracy.
>
> **Q3:** "Insufficient comparison with other works like [1]."
>
> **A3:** We appreciate your emphasis on a more comprehensive comparison with other works, especially the one mentioned [1]. In response, we conducted further experiments comparing the two methods on ResNet-18 with CIFAR-10 and MNIST. Our results, which are detailed below, reveal that the maximum memory consumption of [1] aligns with BP. However, a direct comparison of accuracy is not entirely equitable, as [1] employs a hybrid approach that combines Forward-only and backpropagation methods. To provide a more nuanced comparison, we also adjusted the ratio of BP-pretrained weights (counted from the front) and presented the findings in the ratio column. This approach ensures a more balanced evaluation, reflecting that our method can achieve comparable accuracy with less memory consumption.
>
>    *Accuracy and Memory Consumption on CIFAR-10*
>
> |  ALGORITHM  | MAX MOMORY CONSUMPTION | RATIO | ACC |
> | :----------: | :--------------------: | :---: | :--: |
> |      BP      |          3128          |   -   |  93  |
> |  LoCal+SGD  |          3128          |   -   | 92.8 |
> |     FGD     |          1125          |  0.0  | 45.7 |
> | Async (k=3) |          1187          |  0.0  | 44.8 |
> | Async (k=3) |          1387          |  0.1  | 48.3 |
> | Async (k=3) |          1754          |  0.3  | 64.1 |
> | Async (k=3) |          2247          |  0.5  | 88.3 |
> | Async (k=3) |          2849          |  0.8  | 92.4 |
>
>    *Accuracy and Memory Consumption on MNIST*
>
> |  ALGORITHM  | MAX MOMORY CONSUMPTION | RATIO | ACC |
> | :---------: | :--------------------: | :---: | :--: |
> |     BP     |          3120          |   -   | 98.5 |
> |  LoCal+SGD  |          3120          |   -   | 98.3 |
> |     FGD     |          1120          |  0.1  | 63.2 |
> | Async (k=3) |          1179          |  0.0  | 65.4 |
> | Async (k=3) |          1347          |  0.1  | 69.3 |
> | Async (k=3) |          1788          |  0.3  | 84.9 |
> | Async (k=3) |          2043          |  0.5  | 97.3 |
> | Async (k=3) |          2831          |  0.8  | 98.4 |
>
> **Q4:** "The proposed method has on-par memory consumption with BP based training yet with lower accuracy. The memory cost is crucial for some edge devices."
>
> **A4**: We appreciate your observation regarding the memory consumption of our method compared to BP-based training. Indeed, the gap between BP and AsyncFGD narrows when BP-based methods utilize re-materialization techniques [2, 3], in MLP or RNN-based networks. However, as detailed in Figure 1 of the rebuttal PDF and the Appendix in the original paper, it's essential to highlight that AsyncFGD presents a distinct advantage in convolutional layers, which often constitute the majority of computation in networks designed for edge devices. This advantage arises from the intrinsic duplication of the kernel matrix during the forward pass and the fact that the primary memory consumption in AsyncFGD is for placeholders of random tangents of parameters. These factors contribute to the efficiency of our method in convolutional layers. This makes AsyncFGD particularly suitable for edge devices, where memory cost is a crucial consideration.
>
> **Questions:** Please refer to weaknesses part.
>
> We trust that our detailed responses offer clarity on the concerns you raised. Once again, we extend our gratitude for your constructive feedback.
>
> Warm regards
>
> [1] Krithivasan S, Sen S, Venkataramani S, et al. Accelerating DNN Training Through Selective Localized Learning[J]. Frontiers in Neuroscience, 2022, 15: 759807.
>
> [2] Tianqi Chen, Bing Xu, Chiyuan Zhang, and Carlos Guestrin. Training deep nets with sublinear memory cost. arXiv preprint arXiv:1604.06174, 2016.
>
> [3] Audrunas Gruslys, Rémi Munos, Ivo Danihelka, Marc Lanctot, and Alex Graves. Memory-efficient backpropagation through time. Advances in Neural Information Processing Systems, 29, 2016.

---

> ### Author Response · Authors · 2023-08-17
>
> Dear Reviewer yH2P,
>
> Thank you for your thoughtful review and the time you've invested in our work.
>
> We noticed a positive adjustment in your assessment, and we're truly appreciative of your reconsideration. If there are any further areas where you believe we could refine our work, please feel free to share your thoughts.
>
> Your insights have been invaluable, and we look forward to any additional guidance you may provide.
>
> Best regards

---

### Official Review · Reviewer_Kesv · 2023-07-06

**Soundness:** 3 good
**Presentation:** 3 good
**Contribution:** 2 fair
**Rating:** 5
**Confidence:** 3

**Summary:**

This paper proposes an asynchronous version of the forward gradient descent (FGD) method, in order to alleviate the forward locking exhibited with FGD and enable more efficient implementation. Specifically, AsyncFGD decomposes a network into K decoupled subnetworks that work as a delayed pipeline and enable asynchronous forward propagation.
The authors benchmark their method on an NVIDIA AGX Orin, using a variety of network architectures and datasets. Their method seems to perform better in the fine-tuning setup, where learning rates are small, due to the high variance of forward gradient methods in general.

**Strengths:**

- The paper is well motivated, and tackles an important subject: efficient on-device training
- The authors properly explain their proposed method, the presentation is clear
- The theoretical analysis on the convergence of the proposed method is appreciated
- The authors benchmark their method on actual device and report measured performance

**Weaknesses:**


- The original FGD paper [1] does not report improvements in memory consumption (see Fig 6). I find it interesting that the proposed method does not suffer from the same. I also found that the overall memory discussion a bit lack-luster. It would greatly improve the quality of this paper, and emphasizes the importance of the proposed method, if some details were provided on how memory consumption is computed (or measured?) for all methods (BP, FGD, and AsyncFGD). How bad is the additional memory overhead due to the proposed asynchronous implementation?

- The method has not been benchmarked on big datasets, such as ImageNet. Furthermore, it appears that, much like other forward methods, AsyncFGD cannot scale well to larger problems. This explains why the authors focused on transfer learning applications. This, in my opinion, limits the usefulness of such methods.



[1] https://arxiv.org/pdf/2202.08587.pdf

**Questions:**

- typo in line 104: define
- What is the dataset being evaluated in Fig 3c?
- Why isn't the efficient learning strategy applied to FGD in Table 2?

**Limitations:**

No discussion on limitations.

---

> ### Author Rebuttal · Authors · 2023-08-10
>
> Dear Reviewer Kesv,
>
> Firstly, we would like to express our gratitude for the meticulous review and valuable feedback you provided on our paper. Your insights are instrumental in refining our work, and we have made concerted efforts to address each of the concerns you highlighted.
>
> **Weaknesses:**
>
> **Q1:**: "Why AsyncFGD's memory consumptino is better than that of the origianl paper, how is the memory mesured or computed?"
>
> **A1**: We appreciate you noting the difference in memory consumption between our paper and the original FGD paper [1]. A potential explanation is that the original paper relied on an early beta version of functorch, which may have led to inaccurate measurements. This could explain the perfect alignment of their BP and FGD memory lines. In our study, we used PyTorch's built-in profiler to precisely measure peak memory allocation during optimization. This allowed us to obtain more reliable memory consumption results. Theoretical computation for memory consumption will also be provided in the revised manuscript.
>
> **Q2:** "Worry about scalability to larger datasets."
>
> **A2:** We acknowledge your valid concerns regarding the applicability of AsyncFGD to larger datasets, such as ImageNet. While our initial focus was on specific use cases, we have broadened our experiments to encompass ImageNet in response to your feedback. These findings are detailed in Table 2 of the rebuttal PDF. We recognize a performance gap between AsyncFGD and BP, particularly due to the variance introduced by random perturbation. However, this challenge can be mitigated in online training and adaptive learning scenarios. In such contexts, edge devices initialized with pre-trained weights need to further adapt to different conditions, like auto-driving models, where devices must adapt to specific road conditions using existing knowledge from previously learned experiences, making our method more applicable. Moreover, our explorations into variance reduction with efficient transfer learning are positive. We are also buoyed by parallel efforts of variance reduction for FGD for large scale model and datasets[2].
>
> **Questions:**
>
> 1. **Typo in Line 104**: We apologize for the oversight and have corrected the typo in line 104. We appreciate your attention to details.
> 2. **Unclear Dataset in Fig 3c**: The dataset evaluated in Fig 3c is CIFAR-10. We recognize that this have not been explicitly mentioned in the original manuscript, and we assure that the necessary clarifications will be incorporated in our revised version.
> 3. **Efficient Learning Strategy in Table 2**:
>
>    We apologize for any ambiguity caused by the presentation in Table 2. Our primary objective was to illustrate that AsyncFGD can achieve performance metrics comparable to FGD in transfer learning tasks, even when working with stale parameters. The introduction of the AsyncFGD* column aimed to juxtapose it with AsyncFGD, emphasizing that when paired with efficient transfer learning, our method's accuracy can be further bolstered due to diminished variance. Hence, we did not apply the efficient transfer learning strategy to FGD.
>
> In conclusion, we genuinely hope that our clarifications address the concerns you raised. We remain committed to refining our work based on your invaluable feedback and are confident that our revisions will elevate the quality of our paper. Once again, we extend our heartfelt thanks for your constructive insights.
>
> Warm regards
>
> [1] [https://arxiv.org/pdf/2202.08587.pdf](https://arxiv.org/pdf/2202.08587.pdf)
>
> [2] Ren M, Kornblith S, Liao R, et al. Scaling forward gradient with local losses[J]. arXiv preprint arXiv:2210.03310, 2022.

---

> > ### Comment · Reviewer_Kesv · 2023-08-22
> >
> > Thank you for the detailed response. Regarding the memory footprint, I believe both the measured and theoretical memory consumption of the proposed method (and the BP baseline) should be discussed to have a more interesting and reliable discussion. This will greatly improve the quality of the manuscript. Regarding the scalability issue, it still seems that the proposed method falls short compared to BP. This should be highlighted as a potential limitation of the work.

---

> > > ### Author Response · Authors · 2023-08-22
> > > **Many thanks**
> > >
> > > Dear Reviewer,
> > >
> > > We sincerely appreciate your thoughtful insights regarding memory footprint and scalability. Your suggestion to explore both measured and theoretical memory consumption will undoubtedly enrich the manuscript. We also recognize the importance of clearly highlighting the scalability issue as a potential limitation. Rest assured, these valuable points will be carefully addressed in our revisions.
> > >
> > > With heartfelt thanks

---

> ### Author Response · Authors · 2023-08-18
>
> Dear Reviewer Kesv,
>
> We hope our rebuttal has successfully addressed your concerns. Your insights have been instrumental in enhancing our work, and we are truly grateful for your thoughtful review.
>
> If you have any further concerns or suggestions, please know that we welcome them wholeheartedly. As the deadline for the author-reviewer discussion is approaching, we would appreciate any additional feedback you may have at your earliest convenience.
>
> Thank you once again for your valuable contribution to our research. We look forward to hearing from you.
>
> Best regards

---

> > ### Author Response · Authors · 2023-08-21
> >
> > Dear Reviewer Kesv,
> >
> > Thank you for your insightful and valuable comments. As we approach the final stages of author-reviewer discussion, we want to ensure that our responses have fully addressed your concerns. Your expertise has greatly contributed to our work, and we are committed to incorporating your suggestions in the best possible manner. If there are any points that still require clarification or further refinement, please don't hesitate to let us know. Your continued guidance is deeply appreciated. Once again, thank you for your dedication, assistance, and the positive impact you have made on our research.
> >
> > With sincere gratitude

---

### Official Review · Reviewer_empf · 2023-07-08

**Soundness:** 3 good
**Presentation:** 3 good
**Contribution:** 3 good
**Rating:** 7
**Confidence:** 3

**Summary:**

This paper proposes a novel forward gradient descent method named AsyncFGD to decouple dependencies between layers and thus maximize parallel computation. The authors demonstrate that their method can reduce memory consumption and enhance hardware efficiency through empirical evaluations on AGX Orin.

**Strengths:**

The proposed method of using asynchronized forward gradient descent is interesting and effective, and the authors provide both theoretical analysis and empirical verification, where the latter includes both accuracy and efficiency. The authors also provided extensive analysis and detailed setup.

**Weaknesses:**

One potential issue is that it is not very clear if this work is very suitable for a machine learning venue, as a large portion of the paper are talking about architecture- or system-level issues. Also, from Table 1, it seems the proposed method is not able to achieve consistently better results,like the sDFA or DFA can be better for MNIST, among others.

**Questions:**

Could this method also improve the energy efficiency? Could there be some comparison, even if basic analysis?

**Limitations:**

This paper did not talk about limitations. One potential issue might be related to the environment issue, like carbon emission of the proposed method compared to conventional ones.

---

> ### Author Rebuttal · Authors · 2023-08-10
>
> Dear Reviewer empf,
>
> We sincerely thank you for your comprehensive review and the constructive feedback you provided. Your insights are invaluable, and we have made efforts to address each of the concerns and questions you raised.
>
> **Weaknesses:**
>
> **Q1:** "Unclear Relevance to Machine Learning Venue."
>
> **A1:** You highlighted that a significant portion of our paper focuses on architecture- or system-level issues, which might seem tangential to a machine learning venue. We understand this perspective. However, the system we are exploring possesses unique properties intrinsically tied to Machine Learning. For instance, the layer-wise dependency in each iteration and the iteratively based workload are not typical of arbitrary system processes. Moreover, the strategies we employed to loosen the dependencies among workers are fundamentally rooted in ensuring the convergence of a Machine Learning Algorithm, as evidenced by our comprehensive proof, thus also indicating that we are trying to acclerate FGD from a more algorithmetic perspective. Nevertheless, we acknowledge your feedback and will strive to align our discussions more closely with the machine learning context in the revised manuscript.
>
> **Q2:** "Inconsistency of Advantageous Results of AsyncFGD in Table 1."
>
> **A2:** We value your observation of the advantage of AsyncFGD over other algorithms. To clarity, the target of training on edge device typically not in a train-from-strach, but a online learning fashion, with previous experince and incoming samples from conditions where the device operates. Therefore, our initial Table 1 results may not fully capture AsyncFGD's strengths. Additional experiments simulating online learning offer a more fitting comparison, demonstrating that AsyncFGD's unbiased gradient estimation surpasses other BP-free algorithms relying on random feedback weights.
>
> **Questions:**
>
> 1. **Could this method also improve the energy efficiency? Could there be some comparison, even if basic analysis?**
>
>    You raised an insightful query regarding our method's potential to enhance energy efficiency. We believe that online learning on edge devices, where a combination of previous experience (weights) and incoming samples are processed, is an important implementation scenario for both on-device learning and our AsyncFGD. In such a context, our method offers efficiency by requiring fewer computations per input sample. Specifically, AsyncFGD needs only 2 GEMM (General Matrix Multiply) operations per layer—one for the forward pass and the other for the Jacobian-vector product (JVP)—compared to 3 for BP. This reduction in total GEMMs per input sample could translate into potential energy savings. We recognize the importance of this aspect and will include a more detailed analysis, along with benchmarks on different platforms, in the revised version of our paper.
> 2. **One potential issue might be related to the environment issue, like carbon emission of the proposed method compared to conventional ones.**
>
>    You raise an excellent point about potential environmental impacts like carbon emissions. Analyzing this aspect is indeed complex, as many factors contribute to the overall environmental footprint. On one hand, AsyncFGD might produce more emissions due to its emphasis on maximizing worker utilization, extra context switching, and random tangent regeneration. On the other hand, leaving workers idle and prolonging training can also waste energy, contributing to extra emissions. Thus, there exists a nuanced tradeoff between emissions from longer training time and emissions from additional computations. To provide a comprehensive understanding, we will conduct benchmarks to compare energy consumption and emissions for AsyncFGD against conventional methods across various scenarios. This analysis will enable us to assess the full environmental impact of our method. We appreciate your valuable suggestion to examine this critical aspect, and we are committed to addressing it in our revised manuscript.
>
>    We trust that our responses address the concerns and questions you raised. We are committed to refining our work based on your feedback, confident that the revisions will bolster the quality of our paper. Once again, we extend our gratitude for your constructive comments.
>
> Warm regards

---

### Author Rebuttal · Authors · 2023-08-10

### Sincere Gratitude
Dear All Reviewers and Program Chairs of NeurIPS 2023,

We would like to extend our heartfelt gratitude to each of you for the time, effort, and expertise you have invested in reviewing our manuscript. Your constructive feedback, insightful comments, and valuable suggestions have been instrumental in identifying areas for improvement and refinement.

We have taken all your comments into careful consideration and have made corresponding revisions to address the concerns and enhance the quality of our work. We believe that these changes not only clarify our contributions but also strengthen the overall impact and relevance of our research.

We sincerely appreciate the opportunity to engage in this collaborative process and look forward to any further feedback you may have. Your dedication to maintaining the rigor and integrity of the scientific process is truly commendable, and we are honored to be part of this scholarly community.

Thank you once again for your thoughtful review.

Warm regards

### Remaining Rebuttal
#### Remaining Rebuttal for Reviewer bhPv:

**Q5:** "Section 6.4 unclear for AsyncFGD* accuracy, Async* falls short of BP-baseline."

**A5**: The enhanced accuracy of AsyncFGD* is a result of our efficient transfer learning strategy, which effectively reduces variance. Historically, we've observed that FGD suffer more from variance than optimizing on the subset of parameters. While our method's accuracy might trail BP, we have acknowledged this limitation in our conclusion. We are also optimistic about ongoing efforts to further reduce the variance of forward gradient [6].

**Q6:**: "Section 6.5 lacks memory footprint comparison; how does K affect it?"

**A6:** In response to your query, we conducted further experiments. As detailed in Table 4 of the rebuttal PDF, our findings reveal a 30% increase in memory usage when $K=4$. These results emphasize the importance of carefully selecting the value of $K$, considering both memory constraints and performance goals. Please refer to the rebuttal PDF for a comprehensive overview.

**Questions:**

1. **Missing Content in Section 6.1:** Our apologies for the oversight. We inadvertently added a redundant subsection 6.1.
2. **Clarifications on Sections 6.5 and 6.6**: We appreciate your questions regarding Sections 6.5 and 6.6. We have answered them above.
3. **Proofreading and Typos**: We apologize for any oversight in proofreading. We have taken your feedback into account and have thoroughly reviewed the paper to correct any typos.

We hope our responses address your concerns adequately. We're committed to refining our work based on your feedback and believe that the revisions will enhance the paper's quality. Once again, thank you for your constructive comments.

Warm regards

[1] Rezk N M, Nordström T, Ul-Abdin Z. Shrink and eliminate: A study of post-training quantization and repeated operations elimination in RNN models[J]. Information, 2022, 13(4): 176.

[2] Pellegrini L, Lomonaco V, Graffieti G, et al. Continual learning at the edge: Real-time training on smartphone devices[J]. arXiv preprint arXiv:2105.13127, 2021.

[3] Hagenaars J, Paredes-Vallés F, De Croon G. Self-supervised learning of event-based optical flow with spiking neural networks[J]. Advances in Neural Information Processing Systems, 2021, 34: 7167-7179.

[4] Schaefer S, Gehrig D, Scaramuzza D. Aegnn: Asynchronous event-based graph neural networks[C]. Proceedings of the IEEE/CVF conference on computer vision and pattern recognition. 2022: 12371-12381.

[5] Pan Y, Cheng C A, Saigol K, et al. Agile autonomous driving using end-to-end deep imitation learning[J]. arXiv preprint arXiv:1709.07174, 2017.

[6] Ren M, Kornblith S, Liao R, et al. Scaling forward gradient with local losses[J]. arXiv preprint arXiv:2210.03310, 2022.

[7] [DRIVE Hyperion Autonomous Vehicle Development Platform | NVIDIA Developer](https://developer.nvidia.com/drive/hyperion)

[8] Nøkland A. Direct feedback alignment provides learning in deep neural networks[J]. Advances in neural information processing systems, 2016, 29.

[9] Frenkel C, Lefebvre M, Bol D. Learning without feedback: Fixed random learning signals allow for feedforward training of deep neural networks[J]. Frontiers in neuroscience, 2021, 15: 629892.

### Additional PDF
We have included a PDF file containing the majority of the results from our additional experiments. This supplementary material offers an extended view that contributes to a more comprehensive understanding of our research.

---

### Author Response · Authors · 2023-08-19

Dear All Reviewers,

Allow us to extend our heartfelt gratitude for the time, effort, and expertise you have invested in reviewing our work. Your thoughtful insights have been invaluable in guiding our revisions, and we are deeply appreciative of your contributions.

As the deadline for the author-reviewer discussion is fast approaching, we wish to ensure that we have addressed all your concerns comprehensively. Should there be any lingering questions or areas that require further refinement, we kindly request your guidance. Your feedback is essential, and we stand ready to make any adjustments necessary to meet your expectations.

Once again, we thank you for your significant role in enhancing our research. We eagerly await any additional comments or suggestions you may have, and we assure you of our prompt attention to your feedback.

With sincere thanks

---

### Decision · Program_Chairs · 2023-09-21

**Decision:**

Accept (poster)

**Comment:**

This paper introduces AsyncFGD, an asynchronous adaptation of forward gradient descent aimed at mitigating memory constraints in on-device learning. Supported by robust theoretical analysis and empirical tests on cutting-edge hardware, AsyncFGD succeeds in both minimizing memory usage and enhancing hardware efficiency. After a thorough review, there's a consensus among reviewers to accept the paper. Although I noted a significant accuracy gap in the ImageNet results, the innovative development of an asynchronous algorithm for scalability is an exciting direction. I suspect that some limitations might stem from the nature of forward gradient descent itself, and it would enrich the paper to discuss the differences between backpropagation and forward gradient in the related work section. Given the paper's novel approach and strong justification, I recommend its acceptance.